# The Metabolic Syndrome, a Human Disease

**DOI:** 10.3390/ijms25042251

**Published:** 2024-02-13

**Authors:** Marià Alemany

**Affiliations:** Faculty of Biology, Universitat de Barcelona, 08028 Barcelona, Catalonia, Spain; malemany@ub.edu

**Keywords:** metabolic syndrome, inflammation, energy partition, adipose organ, connective tissue, dietary nutrient handling, testosterone, β estradiol, insulin, protein-fueled anaplerosis

## Abstract

This review focuses on the question of metabolic syndrome (MS) being a complex, but essentially monophyletic, galaxy of associated diseases/disorders, or just a syndrome of related but rather independent pathologies. The human nature of MS (its exceptionality in Nature and its close interdependence with human action and evolution) is presented and discussed. The text also describes the close interdependence of its components, with special emphasis on the description of their interrelations (including their syndromic development and recruitment), as well as their consequences upon energy handling and partition. The main theories on MS’s origin and development are presented in relation to hepatic steatosis, type 2 diabetes, and obesity, but encompass most of the MS components described so far. The differential effects of sex and its biological consequences are considered under the light of human social needs and evolution, which are also directly related to MS epidemiology, severity, and relations with senescence. The triggering and maintenance factors of MS are discussed, with especial emphasis on inflammation, a complex process affecting different levels of organization and which is a critical element for MS development. Inflammation is also related to the operation of connective tissue (including the adipose organ) and the widely studied and acknowledged influence of diet. The role of diet composition, including the transcendence of the anaplerotic maintenance of the Krebs cycle from dietary amino acid supply (and its timing), is developed in the context of testosterone and β-estradiol control of the insulin-glycaemia hepatic core system of carbohydrate-triacylglycerol energy handling. The high probability of MS acting as a unique complex biological control system (essentially monophyletic) is presented, together with additional perspectives/considerations on the treatment of this ‘very’ human disease.

## 1. Introduction

### 1.1. The Human Nature of MS

The wide group of diseases constituting the metabolic syndrome (MS) receives this name because it is assumed to be constituted by a cluster of different synergic pathologies [1], i.e., not being a single disease/disorder. Nevertheless, most of them are related and coordinated, in spite of assumedly being different as to etiology, development, manifestations, and mechanisms [2].

The well-known and independently described disorders/diseases of MS (which are often diagnosed and even treated separately) point to a polyphyletic origin for MS, conjointly affecting patients in varying proportions of development and/or degree of intensity [3,4]. However, the common external (environmental) and internal (metabolic milieu) conditions promoting their appearance relate to evident roots in metabolic dysregulation processes, largely interacting with diet and lifestyle [5]. The syndrome develops with age, following a concatenated branching of prodromal symptoms that progressively give way to a massive intertwined web of disorders running along together for years [6].

The main distinguishing characteristic of MS is, perhaps, its “human” nature. The MS is a pathology affecting growing portions of the world’s population, and includes highly common disorders related to ‘civilization’ (essentially of metabolic origin): ample (and rich) food availability, sedentary habits (because of minimized physical exercise), prolonged lifespan (due in part to the decreased incidence of infective diseases), and an additional protection by strong social/psychological structures [7]. These conditions are superimposable to the benefits of human social evolution and protection, but also to new challenges of knowledge, behavior, and social organization, which constitute an additional novelty in the evolution of concepts such as food security [8].

Some conditions, unique to the humans, have been *transmitted* to animals living under the human aegis: exploited as muscular machines for work and transport, for food and other products, and even (probably the group with higher incidence of MS-like pathologies) used as pets [9]. They are affected—as humans are—by the same pathologic mechanisms of human MS; establishing, this way, a marked difference with the diseases (alone or grouped) that the same animal species may suffer under free-range (wild) conditions. The watchfulness, ‘protection’ (dominion?) and ease of food availability to these animals may provide them with some advantages in lifespan (always depending on the whims/needs of their human herders/proprietors), but they are also exposed to the curse of an also fully human disease (or cluster of diseases), the MS [3,5]. Thus, in this sense, MS is a human-promoted, human-specific, and fully human terminal condition.

Nevertheless, the main subjects of present-day MS expansion are humans. This progressively massive disorder of energy-handling homeostasis affects, already, a large (and growing) proportion of humankind. MS cannot be defined as an infectious disease, but constitutes a truly epidemic health threat [10], affecting predictable subsets of individuals, with a fairly well established onset, course, duration, complications, and (premature) death, despite its inherent variety in pathologic components and depth of affectation.

MS can be considered a consequence of the accelerated evolution of humans’ distorting footprint on Earth, which is right now (in fact, almost out of control) seriously limiting the well-being, and even survival, of a growing portion of humans, and is clearly a consequence of the success of our species in sustaining its disproportioned overpopulation and pilfering of resources at the expense of biological equilibrium on Earth.

This prospective review is focused on the analysis of the biological basis of the MS cluster of pathologies, their close interrelationships and co-dependence, and, especially, their ordinal development tree, hinting at an essentially unique (and human) conjunction of causes eliciting their appearance and development.

### 1.2. The Complexity of MS

The slow process towards the identification of MS as a widely extended cluster of symptoms spun, initially, from the observation of the common association of known diseases and disorders of maturity; the synchronic parallel development of a few important metabolic disturbances expanded to devastating dysmetabolic generalization at the individual level, and up to predictable epidemic conditions in social terms. The identification, concretion, and acceptance of this novel set of syndromic pathologies took considerable time and effort [5,11,12,13]. The first agreed upon main disorders of the syndrome: diabetes, obesity, hyperlipidemia, and arterial hypertension (AHT), affecting (together) the same patients [14,15,16], have been expanded to include practically all forms of inflammation and an almost complete recollection of age and lifestyle-related pathologic traits, universally affecting body systems and functions [17,18].

The collective work of defining, understanding, and placing MS in our medical-knowledge framework has required extensive studies, analyses, and insight to search for a common cause or shared pathogenic path; or, either, to explain the reasons for coincidence (and cooperative effects) of the intervening diseases which are often independently well defined. The initial search for a common cause has been subsumed—in practical terms—in the ‘syndrome’ (literally ‘running together’) concept, which is well extended and often used as a way to define this complex and variable pathognomonic entity [17,18,19,20].

The considerable, and recent, advances in our knowledge of the causes and regulation of the altered-metabolism-based diseases constituting the core of MS have provided considerable insight into their nature and the additive/complementary effects within MS [21,22,23,24,25,26]. Despite the considerable number of ‘syndromic’ traits involved in MS, a careful and systematic analysis of the pathogeny of most of these disorders running together has shown that they share a fairly common number of global mechanisms provoking their surge: inflammation [18,27,28]. At present, the definition of MS as a condition of sustained *low-key inflammation* is being used by a growing number of physicians and scientists [18,27,29,30]. Nevertheless, a systematic analysis of their possible conjoint interrelationships, affecting the growing number of candidate syndromic components of MS, has not yet been completed.

In addition to an ‘internal cause’ for the appearance and development of MS, ultimately based on a deep alteration in the regulation of energy partition and handling, there are quite important additional intervening factors that elicit, develop, and modulate MS in each individual affected by it:(a)*The human-made environment:* the direct relationship of MS incidence with the highly evolved/refined social control conditions (safety, nutrition, reproduction, environmental dominance) that define contemporary humans. The MS can be easily recognized in human-modulated animals and experimental models; for them, at least, MS is a human-transmitted or caused disorder [31,32].(b)*Transmissible predisposition:* it is based on the existing, proven but insufficiently defined genetic [33,34,35] and/or epigenetic [36,37] conditioning of MS development. Most of the MS syndromic diseases are known to be gene-related, at least partially [38,39,40,41].(c)*Socio-epidemiologic factors:* developed in parallel to the growing human control of the environment, which allows for the setting and maintenance of an MS-favoring complex [42,43], but also by food insecurity [44].(d)*Sex-related timing of MS development:* there are differences in the endocrine makeup of MS in adult women and men [45,46,47]. Despite MS affecting proportionally more women than men, the risks implied for mature men seem higher than for women, up to old age [48].(e)*Senescence, the biological transition from maturity to old age*: this process is somehow accelerated by MS, essentially along the lines of inflammation, global metabolic control of substrate partition, and overall physiological activity [49,50,51]. Both paths and effects have been found to be shared by MS and senescence [52,53].(f)*The effect of diet and lifestyle on MS emergence and development:* closely related to type 2 diabetes and obesity (and the coordinated, and derived, cloud of disorders constituting MS) [54,55,56,57].

This list of factors stresses the peculiarity of MS in the context of most occurring pathologies. It is a human-specific and, eventually, human activity-caused pathology, with a defined developmental pattern linked to sex, age, lifestyle/diet, and social heritage environment. There is an inescapable fact in MS being a chronic disease: it is, essentially, a maturity-linked and maturity-onset (additionally progressing with age) set of related disorders that decrease the quality of life and well-being [58]. And which, eventually, shorten our lifespan [59,60,61]. Socio-economic status is a harbinger of increased incidence of MS [62,63,64] but it is, also, a socially altering epidemic with important incidence in the global economy and social structures because of both productivity loss and the growing consumption of health maintenance-related resources [65,66].

The MS has been considered to be a consequence of ‘modern civilization’, i.e., the current culmination of the social and industrious development of humans prevailing over many (infective, parasitic) diseases, making available abundant and nutrient-rich food, lowering their physical workload, and including the very human peculiarity of dying from disease, accident, or age-related disorders, (also including human-generated violence), but seldom being eaten alive as a prey, which is the main cause of death for almost any living creature ‘in the wild’. The ‘common animal’ type of sustained life-threatening stress for avoiding sudden death by a predator has been largely substituted by other types (often caused by social factors) of chronic stress [67,68].

## 2. Pathologic Traits of the MS Cluster

### 2.1. Components of the MS

The MS has been postulated to emanate from a main core disorder, of which intensity increases with time and which spurns complications, pathologic relationships, and consequences, or else prepares the way for the development of additional pathologies taking advantage of the metabolic trail created [69,70,71,72]. A key factor against the monophyletic origin of MS is the common occurrence of many of its main defining disorders as independent medical entities. This approach was fully justified by the “*X syndrome*” or the “*deadly quartet*” [14,73] early definitions, which were described thanks to the knowledge and insight of their defining scientists and physicians. In fact, these efforts for identification and definition were justified by the epidemic proportions of the problem and the urgent need for effective treatments, almost from scratch, confronted with the power and novelty of the ‘association’ of already known diseases. The high degree of superposition of type 2 diabetes and obesity also favored using (for a time) the concept ‘diabesity’ to define this dual synergic (and syndromic) condition [74,75,76]; however, obesity has been largely (and inadequately) attributed to *just* ‘excessive energy intake’, which, consequently, generates an ‘energy imbalance’ that translates into an inordinate accumulation of fat [77], whilst diabetes had a better-known (and assumedly “different”) explanation based on glucose/insulin interactions. The lipid-handling disorders and cardiovascular diseases were deemed more complex because not everything in them could be related to insulin function and control (including the elusive primeval causes of obesity) [4,78].

Despite the fact that many MS disorders can have an ‘*independent*’ development, as well as relatively ‘*independent*’ intensity, pattern and development within the overall context of the syndrome, clinical diagnosis is often obscured by the differences between groups of humans and the florid symptomatology of full-fledged MS (e.g., overweight vs. obesity; prediabetes vs. diabetes; high blood pressure vs. AHT) [79,80,81]. This is even more patent in the case of obesity, a main component of MS which is more easily detected and, thus, diagnosed often before MS [82]. The meaning of fixed criteria may be even less insightful when the comparisons are not made solely from situations with established diagnoses but from gradual analyses of the constellation of disorders over time. Evidently, this does not explain the predominance of one disorder or another within a given MS patient (i.e., AHT or diabetic-/sarcopenic-heart, fostering heart failure, or sleep apnea driving to arrhythmia-related heart failure) but helps disperse the medical focus of attention.

The adscription of known diseases to established different medical specialties does not help the conjoint analysis of a syndrome and the search for a common pathologic path. Evidently, obesity (temporary or established) may be promoted by a profound alteration in diet composition, diabetes, altered nutrient partition, hypothyroidism, inactivity, depression, lactation, stress, bulimia, or iatrogenic/toxic exposure, but also by genetic, epigenetic, ethnic, or social environment factors [82,83]. However, in all cases, there is an inordinate ectopic accumulation of fat, whilst (at least) lipid and carbohydrate metabolism are altered, and insulin function is affected [84,85].

The large amount of data available suggest that diet plays a critical role in MS’s appearance and development [86]. The incidences of sarcopenia (and frailty) are becoming more common [87], and the affectations of insulin [85,88,89,90,91], androgen [92,93,94], estrogen [95,96], and glucocorticoids [97,98] have been found to be generalized. Similarly, the metabolism of carbohydrates and lipids is deeply altered [99,100], but also that of proteins/amino acids [101]. The effects of MS seem to be most crippling on the liver [102,103,104], heart, and vessels [105,106,107,108], as well as the endocrine (such as gonads and adrenals) [109,110] and immune [111,112] systems, but practically every organ and tissue is affected: the brain [113,114], kidneys [108,115], skeletal muscle [116,117], connective tissue [117,118] (adipose [119,120]), lung and respiration [121,122], bone and cartilage [123,124], skin [125,126], including our different microbiota niches [127,128,129]. The degree of intensity of affectation of organs/systems may vary, but the extension of the consequences (i.e., overall endocrine and defense milieu, redox state [130,131], substrate partition [132,133], and handling) affect the whole body.

### 2.2. MS Main Paths and Mechanisms

The metabolic core of MS incorporates a relatively short list of key terms: type 2 diabetes/insulin resistance [134,135,136,137], obesity (or abnormal, excess, ectopic fat deposition) [138,139,140], and hyperlipidemia [141,142,143], as well as their derived consequences (or parallelisms): liver steatosis [88,144,145,146], heart and circulation alterations [107,147,148,149], and generalized substrate-handling driving to metabolic dysfunction [150,151]. This process includes, essentially, altered insulin, but also estradiol, testosterone, and glucocorticoid function [152,153], as well as dietary changes, such as excess energy intake [154,155], excess lipid or carbohydrate, and inability to maintain energy homeostasis [5,156].

The control of energy balance belongs, initially, to the nervous system, which, through the control of the status of energy balance [155,157,158], acts via regulation of appetite [159,160,161,162] and, especially, controlling the main hormones implicated in the process: insulin, glucocorticoids (GC), testosterone (T), and 17β-estradiol (E2) [159,163,164,165]. Insulin plays a central operative role, and its action is regulated by the pancreas, intestine, liver, and the cited steroid hormones, acting as the main direct regulator of glucose availability and the handler (largely oxidation or energy storage as lipid) of this primary substrate [157,158,166].

Thus, a substrate/energy path of
**food → nutrients → partition:**→ to oxidation (yielding NADH, ATP, heat)
→ or energy storage (as fat, glycogen)

and a parallel regulatory axis of
**brain** (*food intake, endocrine control*) → **hormones** (*insulin, T, E2, GC*) → **nutrient partition**control the response to eventual situations of excessive availability of energy substrates. Their combined functions determine the shifts in energy balance modulators that maintain homeostasis, and/or counter the deviations towards unwanted energy mishandling driving to its accrual (liver steatosis, obesity, diabetes, hyperlipidemia, etc.) that characterize MS [167]. Most of the MS-related conditions, listed in Table 1, can be traced to this nuclear metabolic core:
*food intake → digestion/assimilation of food → splanchnic triage and partition of**nutrients → oxidation for energy of substrates and/or storage, of the energy balance.*

A brief description of the main known pathogenic mechanisms of MS provides additional information to this succinct scheme. These diseases or disorders include the following:(a)*Hepatic steatosis, with loss of hepatic function, disorders of redox status, and energy partition*.Interrelated with disorders of lipid synthesis disposal, transport, and deposition, such as obesity and hyperlipidemia [156,168], as well as alterations in supporting metabolism, such as that of purines/urate [169,170].The liver is the main site for digested substrate-handling; liver damage results in the rapid extension of a wave of altered functionality to peripheral tissues, eventually affecting the regulation of whole systems. Liver steatosis (i.e., hepatic parenchyma accumulation of unused triacylglycerols –TAG– and altered two-carbon catabolites –2C– handling), also known as NAFLD (non-alcoholic fatty liver disease), has been proposed as the mainstay (and/or core initial metabolic disorder) of MS [104,171] because it affects the critical node of energy triage and partition between intestine/diet and the systemic circulation [133]. But also because, in the core, NAFLD is a consequence of altered insulin system function, resulting in the inability to dispose of excess 2C energy at the main point of intersection of human systems with the main body (gut) microbiota. This results in an excessive export of TAG-laden lipoproteins that the liver secretes because it cannot handle them [133,152,168]. Chronic inflammation of the liver can induce a domino effect on many other homeostatic systems, giving way to the appearance of known MS component complications/pathologies [172].(b)*Insulin resistance, intolerance of glucose, and type 2 diabetes*.Related to hepatic function, steroid hormone disorders, and energy partition, especially the handling of carbohydrates and its regulation [173,174,175], including the deposition of TAGs as energy reserves (or ectopic deposition of TAGs anywhere as a way to dispose of excess of unneeded 2C) [152,176,177].Related to disorders caused by the lack of effective insulin function [178,179], including blood microcirculation and renal function [180,181,182,183].(c)*Hypoandrogenism, plus low estrogen levels and disordered gonadal and adrenal estrogen and androgen synthesis*.Related to brain function, genetic inheritance of metabolic alterations, and inherited patterns and functional blueprints for metabolic responses to alterations in normalcy [184,185,186], as well as the adjustment to biological/metabolic rhythms [187] and bone protection [188].Related to the regulation of insulin handling of glucose [189,190,191], control of 3C (three-carbon gluconeogenesis-related substrates) to 2C conversion and disposal of 2C for energy in the mitochondria [192,193,194], as well as keeping redox status and energy partition [133,195,196].Related to protein preservation and accrual [197,198,199], as well as oxidation of their amino acids as energy/anaplerotic/biosynthetic substrates.Related to hypogonadism (and sex, reproductive functions [200,201,202], behavior, and interpersonal relationships).(d)*Disorders of corticosteroid synthesis and function and adrenal redox state*.Affecting rhythms, behavior, and additional regulation of hormones, but also the differentiation and growth of cells and tissues [203,204]. Corticosteroids play a critical role in the defense system [205] by modulating and adjusting the immune response [206,207], both downplaying the immune response [208,209] but enhancing its effectivity [210,211,212]. Their function is necessary to prevent/correct the damages caused by inflammation [213,214,215].Interrelated with all the previous points presented here (a–d) and the next one (e). An important aspect of corticosteroids is their regulation of blood vessel reactivity, affecting blood flow and pressure [203,216], largely in contraposition to DHEA (dehydroepiandrosterone) [217,218]; interacting with T (and other androgens), E2 [110,188,219], and insulin [220,221]; and even acting in coordination with catecholamines [222,223].Corticosteroids also show a critical effect on behavior, in conjunction with other steroid hormones [224,225], including appetite (via a myriad of additional cytokines, non-coding RNAs, and other protein factors). The relationship with neural transmission and the modulation of the complex nature of depression has been also acknowledged, but the mechanisms have not yet been fully established [226,227,228].(e)*Disorders of cell and tissue matrix, interrelationships, and interstitial space/tissue.*Related to energy partition and availability, splanchnic–peripheral relationships, composition of interstitial/connective tissue [229,230,231], and to the immediate relationships with ectopic fat deposition and obesity [232].Related to disorders of defense systems (immune, apoptotic, and regenerative processes) maintaining homeostasis and tissue defense [233,234]. Direct relationship with the preceding group (d) of syndromic disorders (e.g., corticosteroids).(f)*Indirect relationship with the microbiota*, as well as defense reactions to infection and the effect of alien (not necessarily active or toxic) agents.Consequences of immune overreaction such as asthma, allergy [235,236], and dermatitis, including the classical “four Ds” of deficit (pellagra) or acute insult affecting, essentially, cells derived from the ectodermic layer: dermatitis (skin), diarrhea (digestive canal), dementia (brain, nervous system), and death (global irrecoverable disorders or disproportionate reactions to them). This is a relatively localized response (often autolytic).(g)*The probability of additional ties with rheumatic diseases (*via* tissue redox and immune homeostasis)*.This part, including metabolic stress, is currently being studied at speed, probably prompting its incorporation into this group of rheumatic diseases such as osteoarthritis [237,238,239], but also psoriasis [125,126,240], as two representative syndromic effects on bone and (primarily) the skin, both of developmental epithelial origin.(h)*Probable incorporation of the ‘metabolism’ of some essential minerals, such as Mg, Zn, Cu, and, especially, Fe*.Related to liver steatosis [241,242], to coagulopathies and alterations in hematopoiesis [243,244], as well as to the regulation of apoptosis and ferroptosis [245,246]; direct implication of metal ions in the regulation of insulin handling of dietary carbohydrates [247,248].(i)*Implication of most of the disorders exposed here with cancer and other altered mechanisms of cell tissue proliferation, growth, and altered control* [249,250].These mechanisms are many and diverse, including the loss of tight metabolic–hormonal control and/or severe nutrient dysfunctions, mainly due to excess/disarray of dietary nutrients (and/or their handling in the gut and liver). Interactions with tumor tissue may result in the opportunistic hijacking of pathways [251,252,253], organelles [254,255], or signals to promote neoplastic growth [256] and, especially, cells: fibroblasts [257,258,259], adipocytes [260,261,262], stem cells [263,264,265], macrophages [266], and other defense-related cells [267]. This is quite a different type of disorder compared with the previous ones but shares its global activation by the syndromic pathologies that constitute MS in its wider definition. In sum, these overlapping mechanisms and actions are directly related to the fracas of the specific cell and tissue mechanisms of defense, and to the evolved facility of some types of cancer (including that provoked by viral or comparable means of infective transmission of the attacking blueprint) to grow as if its nature were simply parasitic [268]. The key point here is not the diverse metabolic approaches taken by the modified cells, but the biological meaning of this complex group of syndromic disorders. The possibility of them forming part of a coordinated culling mechanism as that postulated for the entirety of MS cannot be, right now, ruled out.

### 2.3. Other Aspects of MS

This part does not refer to secondary or simply unknown aspects, but to components of MS that have not been sufficiently studied and/or of which severity and risks are either known (and some medical procedures can be used on them, at least symptomatically), or for which we are yet in a penumbral state of knowledge as to their pathogenicity. Most of them are considered full ‘members’ of MS, but, in other cases, their synergy has not yet been sufficiently clarified, or even ascertained.

(a)*Human behavioral patterns affected by MS*.These encompass the whole spectrum of eating disorders, often accompanied by well-defined behavioral or psychiatric pathologies [269,270]. In most patients, social rejection aggravates their problems, but this reaction is often compounded by other pathologies. Probably, altered behavior is (at least in a significant part) a consequence of the deep changes that MS (enhanced by age) induces in the lifestyle of those affected [271,272]. The current appreciation of this group of alterations is largely derived from these parameters.(b)*Cardiovascular diseases*.They are almost omnipresent in populations with MS, and constitute, perhaps, its most common consequences [273,274,275] and the subject of a few apparently beneficial ‘paradoxes’ [276,277]. They can be summarized by an increased risk of AHT [278,279,280,281], atherosclerosis [282,283], stroke [284,285], arrhythmias [286,287], and heart failure: coronary insufficiency or claudication [288,289], loss of heart function because of damaged tissue or contractile rhythm, signal transmission, or bringing up altered function because of degenerative processes [290,291]. Cardiovascular diseases are a main cause of death and incapacity attributable to MS [107,292], but the total risk of MS remains lower than those of these individual disorders taken independently [293]. The origin of these cardiovascular pathologies is a consequence of distorted metabolic function and regulation, which may result in sarcopenia [294,295], damaged tissue [296,297], altered vessel function [298,299], loss of heartbeat rhythm or dysfunctional signal transmission [300,301], altered blood coagulation [302,303], etc. These disorders constitute a legion of related, coordinated, or secondary actions which have already been presented as part/consequence of the main MS pathological traits. They are indeed a sum of different pathological paths (in the purest syndromic style), but their causes (and treatments) are to be found in the core of the main groups of mechanisms described in Section 2.2.(c)*Mass-dependent problems ‘package’*.Related to consequent, parallel, or causing damages that affect the operation of the circulatory system and, consequently, alter the organ functions: kidney [304,305], retina [306,307], adipose tissue [308,309], and lung insufficiency [310,311]. Pulmonary function is critically affected by MS [312,313,314], in line with generalized cardio-vascular alteration [292,300]. Inflammation in the lung [315] probably enhances the chronicity and severity of infective, toxic, autoimmune, or occlusive diseases of quite different origins, causing fibrosis and irreversible loss of function, such as asthma and pulmonary chronic obstructive disease [316,317]. MS also induces a higher lung susceptibility to infection [318,319] and cancer [320].(d)*Common (or not) disorders being ascribed to the MS spectrum, but which mechanisms are not clear fully clarified*.This group includes the classic acanthosis nigricans [321], directly related to obesity and insulin resistance [322,323]. Obstructive sleep apnea is a widely extended disorder, related to obesity [324] or MS [325,326], which has been linked to insulin resistance [327]. Sleep apnea has been postulated as a cause of insulin resistance [328], but the nature of this relationship has not been fully clarified [329]. Sleep apnea and its consequences on adrenergic signaling have also been related to AHT [330] and other cardiovascular effects [331,332]. The induction of atrial fibrillation is a known path for these interactions [301,333,334].(e)*Immediate physiological consequences of disturbed blood circulation, including the loss of heart pumping capacity and the related appearance of edema (or, even, lipedema)*.These are direct physiological dysfunctions/responses to the insult of dysfunctional circulation [335,336]. We can also include here the renal [337,338], respiratory (pulmonary hypertension [339]), eye (glaucoma [337]), or adipose [340] problems caused by altered organ circulation and fluid handling. Most of these effects, despite their importance (and often crippling loss of function), are the consequences of fairly known mechanisms. Unfortunately, this ‘knowledge’ does not mean that the effects can, consequently, be successfully treated, since this depends on the particular nature of the core problem causing the disorder.Obesity often carries with it severe additional consequences, including the loss of mobility [341,342] and misadjusted water/electrolyte handling [343] or thermoregulation [344,345]. Some of these effects can be attributable simply to the effect of the extra body mass to be held, physically supported/carried, and provided with adequate nutrients and oxygen supply through an (also altered) blood flow; a burden additional to the metabolic and regulatory distortions that cause the obesity hypertrophic and/or hyperplastic disordered growth. There is a biological (albeit not fully mechanistic) parallelism with tumor growth, in which the migration of adipocytes and fibroblasts helps nurture and consolidate the neoplastic growth (Section 2.2 (i)).(f)*The expanding field of diet- or regulatory-induced microbiota alterations*.These directly affect the core of MS pathologic interactions and introduce an additional layer of complication to all other MS manifestations [127,346,347], contributing to obscure the basic trends in MS development [348,349] and its chronic cumulative damage [350,351].(g)*In an indirect way, the MS-induced changes in body function have a direct translation into behavior, including society-generalized changes in habits*.This question is in line with the possible interpretation of MS along a ‘planned obsolescence’ hypothesis, requiring—and provoked by—social interrelationships and the requirements of basic social structures [201], which antecedents are even previous to our present-day species. In addition, the often superfluous (when not utterly negative) intents to treat deep metabolic processes using prematurely launched, simplistic, and even non-knowledge-supported procedures to treat the MS spectrum of diseases have deep and long-lasting consequences to our health. The spot (and often aleatory or ‘ideological’) changes in the diet (e.g., ‘zero’, carbohydrate-free, dissociated, lactic, nutrient-restricted diets) may result in further damages to the health, hope, patience, economy, and trust of the patients, despite the huge amounts of resources invested in their apparently proven ‘healthy’ results (and almost nil effectiveness on MS) [352,353]. In this complicated path, there have been considerable setbacks (mostly because too often hope has substituted knowledge and proof). In this list we can include the fracas of sustained hypo-energetic diets, the β_3_-adrenergic agonists’ boom, the hopes put on leptin, ghrelin, and other hopeful miracle treatments… and thousands of compounds, procedures, and guides that, at most, have had only marginal effects on the ‘cure’ of obesity. Most of these failed intents are the fruit of the ‘need’ to treat millions of patients asking for a rapid and effective solution to their crippling health problems. This desire is compounded by the frustration of health professionals and the persistent lack of effective knowledge-based canonic treatments. It is often ‘forgotten’ that the bottom line for this lack of responses lies in the inadequacy of the research conducted and the real interference of corporate interest-focused financing [354,355], but also in of the piecemeal cumbersome system of scientific diffusion and the final control of omnipresent spurious interests [356,357]. We can only hope that this situation will eventually subside when our scientific knowledge (and the unconditioned access to it) reach the critical mass needed for solving a problem practically created by humans and which essentially affects humans: the MS.

Figure 1 shows an schematic summary of this Section, establishing the main relationships between the full-spectrum of MS components, their regulatory interdependence and implication in the core control of energy handling.

## 3. Inflammation

### 3.1. Inflammation, Concept, and Types

Inflammation is assumed to be a main process driving MS; however, this generally accepted (global) idea [18,28,30] is in fact ill defined. “Inflammation” is a clinical term that includes a wide array of situations of altered homeostasis through quite different pathways and mechanisms [86,358]. In MS, inflammation (latu senso) is at the root of many of its collection of disorders [18,359]. The question lies in the lack of a precise univocal definition of inflammation, apt for all instances in which it is used to define reactive processes of biological systems, to externally induced disruptions, to the maintenance of homeostasis.

“Inflammation” is quite an old (and successful) term; it was initially defined by the late-Roman Aulus Cornelius Celsus in his work *De Medicina* [360]. He described inflammation as a marked effect or sensation, localized in tissues, comparable to them “burning” or being “in flames”. He further detailed this condition through four summative effects: *calor* (heat), *rubor* (reddening), *turgor* (turgor, swelling), *dolor* (pain), and a main consequence of them: *functio lesa* (loss of function). This is quite a good definition of tissue alterations due to infection or damage, which is proven by its continued use (and its further extension to more-or-less parallel or comparable situations) fifteen centuries later. However, these simplifications are quite insufficient [361], despite the long series of historical modulations of the concept. *Calor* and *rubor* are, assumedly, the consequence of increased blood flow towards the affected zone; *turgor*, swelling, is the edematous response to capillary fluid extravasation; and *dolor* is a warning signal of functional alteration in the affected zone. This definition of inflammation, even accounting for the mechanisms of capillary wall permeability, increased overall supply of blood, and the generation of pain signals (i.e., through histidine decarboxylase, prostaglandins, and proprioceptive signals), deals only with physiological-level phenomena, meaning that it refers to organs or tissues under insult. Consequently, the use of the term inflammation, without further elaboration, is, thus, not technically fitting to describe metabolic alterations resulting in a deeply altered function of cells and/or their structural components. It is, then, inadequate to use the term to describe (without further qualification) reactive biochemical, cell, or even tissue processes, since the mechanisms implied fully affect different organization levels of organs, tissues, and/or cells. 

The four defining effects of inflammation cannot be easily translated to the consequences of low-key inflammation processes [86,362], or the dysfunctions of endoplasmic reticulum and mitochondria [363,364,365], under the effects of excess energy availability [28,137]. The relationship of MS with inflammation independently affects most of its components despite its global effects [366,367]. These processes occur at a lower structural level, closer to molecules than to global organ function, i.e., than the classical physiology definition described above and of comfortably accepted reference [358,368].

The common signs of inflammation are the consequence of modified or altered molecular processes which regulate the gross physiological mechanisms affected, but the consequences of mitochondrial receptors’ modulation in target tissues cannot be *directly* comparable with the (essentially wound- or infection-driven) clinical inflammatory responses; the processes may show a biological analogy, but the different processes implied are not mechanistically homologous because of the different organization level at which they take place.

There is another key factor required to understand inflammation (at all levels), and it is the marked protective/defensive nature of the mechanisms implied. In classical (or clinical) Celsus-like inflammation, the increase in blood flow brings more oxygen, nutrients, and defense-related cells to the affected site in order to fight against alien interference and, later, promote/allow for recovery and restoration [369]. The interstitial space is optimal for the strategic movement of phagocytic cells, or access by immune-system molecules, because of the possibility of maintaining/restraining all components within a limited space or ‘killing field’ [370,371], corralling inside too the microbes or foreign objects provoking the tissue reaction. The resolution of inflammation often includes the accumulation of debris and its progressive (also phagocytic) elimination [372] up to the recovery of normalcy [373,374]. Comparatively, at the cell/tissue level, this type of response is more constricted, and is largely fought by a keen combination of apoptosis/autophagy and phagocytosis, as well as signaling and direct interactions between molecules [372,375].

Taking into account these caveats, and from a mechanistic and functional point of view, two different intertwined levels of inflammation responses can, at least, be established:Classical or clinical inflammation [CI] (basically that of Celsus). The term ‘Clinical Inflammation’ may be preferable, since its descriptors fit, at the level of organization and context, with common clinical symptoms, with its descriptors (mechanism and function) being interchangeable.‘Biochemical Inflammation’ [BI] is adequate to describe the ‘inflammatory’ phenomena occurring at the molecular and structural cell levels, i.e., the common realm of biochemistry.

The eventual description of BI as ‘Molecular Inflammation’ may also be inadequate, since the molecular level of organization spans from the atomic environment to macromolecule structures, which exists at a lower level of organization than that including cell (and/or extracellular) structures (despite the common use of the lax concept of ‘Molecular Biology’ (and derived terms) to refer, essentially, to a limited set of large complex cell structural systems (from macromolecules upwards), as in its origins, its definition was more circumscribed [376].

Using this overlapping dual organization-level approach, we can retain the ‘successful’ term-concept of inflammation, since the key is, again, the level of organization at which the implied parties and the process interact: agents, targets, theatre, action, and resolution. In any case, emphasizing their implication within the overall ‘biological’ framework of the sum of processes. In CI, the organization level extends from whole body to organ, tissue, and cell, with its mechanisms ranging from systemic to tissue. In parallel, BI spans the structural level from cell to subcellular/molecular level, and the mechanisms are based on molecular-level interactions. Both inflammation types share the critical (central) ‘cell’ structural-level node, albeit through different methodological approaches. The use of ‘inflammation’ to encompass these two different sets of mechanisms is acceptable as a general concept, but also because it is used much too extensively to be changed. In addition, the focus, objective, and defensive alarm-and-destruction sense of inflammation is shared, albeit using, again, quite different (but nevertheless related, complementary, and superimposed) mechanisms.

### 3.2. Tissue-Related Inflammation Mechanisms: CI, Clinical Inflammation

Figure 2 shows a scheme of the three main actions of clinical, Celsus-defined inflammation. The presence of particulate matter (i.e., biological agents, infective cells, spores, protein aggregates, small-size debris, and products of biological strife, damage to cells and tissues), even inert components, and/or biologically active materials/agents are detected by tissue physical or biochemical integrity status sensors [377,378,379]. Their stimulation elicits the local (and/or systemic) release of mediators of inflammation, i.e., by mast cells [380,381,382]. The signaling molecules, such as histamine [383], cytokines [384,385], or interleukins [386,387], add to increased levels of defense system factors such as C-reactive protein [388,389], interferons [390], and other paracrine secretions [391], as well as the rapid changes in the concentration of signaling ions (e.g., calcium, zinc, iron) [392,393], redox state [394,395], action potentials, and pH. The implication of the extracellular matrix in the inflammatory process is critical [396,397], both as a medium to allow for the action of the cells implicated and for their control of regulating agents [398,399]. The array of signals and mediators and the responses provoked are partly tissue-specific and partly generalized or systemic; inflammation is a wide, interactive, complementary, and complex process.

The main responses to the presence of the mediators of inflammation at the body-to-cell structural range [CI] are as follows:(a)*Increased capillary permeability.*This results in an increase in capillary fluid leaking into the interstitial space, swelling the tissue area (i.e., mostly enlarging the interstitial, intercellular, or tissue extracellular spaces) and also allowing for a number of proteins, largely related to defensive recognition, binding, and inactivation of potentially harmful elements, to pass from plasma to tissue. The increase in volume cannot be rapidly corrected because the main way-out of fluid, osmotic uptake in the capillary venous side, is slow and has been altered; furthermore, lymph efflux is also diminished, especially in lymphedema [400,401]. Brain tissue inflammation is peculiar, since the swelling cannot be readily corrected by the slow CSF dynamics but is instead by the long cycles of glymphatic wave-washing [402,403]. Extravasation also allows for the incorporation of coagulation factors to the interstitial extracellular space, which, when and if necessary, clot to prevent hemorrhages and to help maintain the isolated inflamed space [404,405] in which limited, controlled, and largely unconnected efflux helps to prevent the dissemination of the ‘perceived danger’ agents (and the responses to them). The swelling and turgid stasis results in the partial loss of function and homeostatic equilibrium, at least until (if) full working order in the tissue is recovered [375].(b)*Local vasodilation.*It occurs mainly at the arteriole level [406,407], and results in a locally increased capillary blood flow [406,408]. This helps activate the extravasation described in point (a), and the maintenance of turgidity, which also limits the clearance of excess fluid and, thus, helps the maintenance of the inflammatory theatre limits. In addition to raised blood pressure, the increased blood flow results in a higher supply of oxygen and nutrients (including specially required amino acids such as histidine and arginine) to the tissue [409], as well as eliciting a higher temperature than that of the surrounding tissue and even higher than that of the body’s core [410]. The higher temperature, oxygen, and energy supply of substrates increase the metabolic activity within the circumscribed space and also enhance the activity of defense system agents, such as phagocytic and antigen/antibody-related cells [411,412]. Inflammation is often related to fever, an increase in temperature (and BMR) which is centrally controlled and helps to globally activate the defense system, i.e., enhance phagocytosis [413].(c)*Recruiting of phagocytic cells in the inflammation theatre.*This includes the recruitment and selective polarization of macrophages [414,415,416,417] and their chemotactic (or electrophilic) migration from neighboring spaces or tissues [418,419,420], including different types of leukocytes, and, especially, those arriving with the raised inflow of blood [421,422,423]. The cells bind or cross the capillary walls (via diapedesis) in high numbers [424], a process facilitated by their increased abundance in the blood influx to the affected zone. The inflamed tissue draws blood monocytes and converts them into macrophages, but also receives, especially, neutrophilic leukocytes, which fight infections close to cell membranes in massive waves of intense phagocytic activity [425,426] The increased concentration of phagocytes and higher temperature (oxygen plus energy substrates) multiply their eventual activity, killing microorganisms or just embedding any suspicious particulate matter [427,428]. These actions result in the massive accumulation of tissue debris, dead cells full of partly digested microorganisms or other materials, altered matrix components, and inactivated alien structures (i.e., pus), which marks the end of the defensive part of the battle [375].

The inflammation process can be turned down when the signals of an assumed aggression have been eliminated; then, the process of recovery tries to bring the tissue back to normalcy, i.e., the pre-inflammation status [373,374]. This is achieved by: (1) cassation of the detection of danger and release of (i.e., signaling by) mediators of inflammation; (2) the progressive turning-down of hyperemia to normalized tissue blood flow; (3) the restauration of limited capillary permeability with the complete restoration of capillary circulation, lymph efflux, and turnover; (4) fibrinolysis and resorption of clots (if any); (5) restoration of damaged tissue and extracellular matrix (i.e., via fibroblasts); (6) the final elimination of debris, again largely via a slower and more complete phagocytosis process, which essentially results in the complete oxidation of the biological waste material, irrespective of its origin [372]. The process of recovery is helped by the action of trophic cytokines, in part commandeered by central signaling [385,429,430] and the protective effects of androgens [431,432], estrogens [433,434], progesterone [205,435], and DHEA [436,437]. The role of glucocorticoids [438,439] is essential because they provide a needed restraining control over the immune response (in collaboration with androgens) to prevent corticoid overshoot-caused damage [440].

### 3.3. Cell-Related Mechanisms: BI, Biochemical Inflammation

As indicated above, there is a continuum between CI and BI, the latter largely supplying the agents and pathways to implement the physiological changes. This interaction is made fully transversal for immune-related systems, with the whole system of phagocytosis, activation of phagocytic cells, polarization chemotaxis, etc., on one side [417,441], and the immune response on the other [442].

The BI mechanisms can be organized along pathways and subjects of modulation, mainly in the following groups:(a)*Modulation of gene expression because of genetic and epigenetic signals*.Resulting in cascade effects induced by differential expression of agents favoring inflammation or not. There is considerable bibliography on these mechanisms [443,444,445,446], but despite the biological reasoning that stresses their importance, the difficulties inherent to their exposure, tracing, and understanding have prevented most of them being a part of already mainstream knowledge.(b)*Dysoxia-related processes*.Showing alterations in the availability or handling of oxygen. The effects of hypoxia considerably limit the oxidative generation of energy and, thus, the effects extend to all corners of intermediate metabolism [447,448]. Changes in the redox state of cells and cellular systems also condition metabolic pathway shifting and/or modulation of (or disordered) substrate utilization and regulative systems shifts [449,450,451,452,453].(c)*Accumulation of free radicals*.Because of their inordinate production and/or altered mechanisms of suppression [454,455]. Free radicals are especially damaging in the context of membrane polarization and signaling [456,457], as well as in the handling of metallic ions, affecting their implication in membrane transport [458,459].(d)*Alteration in metal ion roles*.In relation to membrane barrier/transport/gate control functions, which modulate the cell compartment distribution of, mainly, calcium [460,461], magnesium [462,463], zinc [464,465,466], and iron [393,467] as part of their signaling/cofactor functions. Evidently, the Na^+^/K^+^ equilibrium is another critical factor, often altered over the development of inflammation [468,469] because of its signaling and transport functions.(e)*Cell cycle modifications*.Elicited by external factors spurning inflammation or internal factors, such as disorders of regulation, substrate energy handling, or genetic/epigenetic modulation [452,470,471]. These mechanisms often include the modulation of cell differentiation (i.e., fibroblast transformation [472,473]) or response to stimuli (i.e., macrophage polarization [416,417]), but also include changes in cell-type distribution in tissues (i.e., in adipose or bone marrow [474,475,476]) or the elimination and deep reconstruction of cell patterns (i.e., such as apoptosis [477], ferroptosis [245], remodeling of adipose tissue [478,479]), changes in mitochondria numbers and functions [480,481], and, especially, the transfer [482,483] and biogenesis of mitochondria [484,485]. This is one of the main cutting-edge areas in cell biology at present, and is probably the main focus of attention on inflammation because of the coincidence of three main vectors: (a) tissue reshaping/remodeling as an adaptation to shifting needs [483,486,487]; (b) adjustments in energy and protein/energy accrual or wasting [488,489,490,491]; and (c) the production of specialized cells, mainly in the orbit of defense systems (immunity, phagocytosis) [492,493].(f)*Control of possible incorporations to the working genome* (eventually including DNA from alien sources).This refers to the Lamarckian adaptations of live human beings—and, especially, of their descendants—to the conditions of the medium and their genetic [494,495] (but mainly epigenetic [496,497]) modulation. The implication of the diverse niches of the human microbiota has extended our horizons to the modulation not only of our cells and tissues, but also to the commensal, symbiotic, and even parasitic cells that do not share our DNA but live within us and are a necessary part of us (as a biocenosis) [498,499]. The transfer and handling of cell organelles (mainly mitochondria, so far) may be contemplated within this context of tissue structure, function, and cell composition shifting [482,500,501,502].(g)*Handling of body (i.e., cells and tissue matrix) protein*.In the complex context of protection/sparing of amino N sparing [503,504], including the retention and reuse of amino nitrogen and, especially, essential amino acid hydrocarbon structures [505,506]. These complex processes occur within the context of omnivorous feeding, which includes the use of protein as a common/usual staple for energy generation. Our bodies need to maintain their total protein mass/pool as fully functional at any time, despite senescence [507,508] and even under the influence of MS [509,510]. This issue has, so far, been poorly studied for obvious methodological constrictions and even within the context of dietary needs, since many “recommendations” are no longer sustained by current knowledge. Nevertheless, there is information available on structural problems in the cell handling of proteins, such as protein folding errors [511,512] and their elimination [513]. There is, also, a growing interest in the metabolic remodeling of intercellular matrix proteins, including the action of some cathepsins [514,515]. Recently, as an example of an expansive analysis of our biocenosis, it has been found that some viruses may affect/interfere with the fine-tuned system of selection of mRNAs to be translated into newly formed proteins [516].(h)*Excess of energy-providing nutrients*.Mammals have evolved to develop successful strategies to endure the harsh (albeit common) conditions of insufficient availability of food [517,518]. Adaptation to food deprivation elicits a rapid, compensatory reduction in energy expenditure [519]; when needed, the use of body reserves of energy substrates is activated [518,520], eventually carrying on the deconstructive oxidation of body structures. All of these actions are centered on the priority of maintaining our essential functions in order to keep surviving by using the metabolic energy remaining in the body as life-sustaining fuel [517,521]. The reverse condition, i.e., “how to cope with an ‘excess’ of energy available” is so rare in any ecosystems’ context that no evolutionary mechanisms have been developed yet to face such a biologically improbable situation [5]. Unfortunately, the biological success of humans has been (so far) overwhelming, and has developed within an extremely short (historical, not biological or evolutionary) period; the maintained availability of food has ceased to be a main problem for a large part of humankind. In fact, a consequence of affluence, the inordinate accumulation of body energy reserves—in practice never to be used, stored in counterproductively large amounts—has become a serious health problem by itself [522,523]. Now, obesity is a main component of MS, a disease of ‘excess’ availability of energy, protein, diet micro-components, etc., but lacking a possible counter-regulatory escalation of the basic and well-regulated energy expenditure system of mammals [524]. This is probably a critical factor for the development (and maintenance) of MS. The question of energy balance, diet, and MS is further developed in Section 4.4.

## 4. Metabolic Regulation-Derived Inflammatory States

### 4.1. Energy Balance

Energy balance establishes the real availability of energy, in different forms, to carry out the complex task of living, centered, from a biological point of view, essentially on ‘surviving’ and reproducing, i.e., the perpetuation of the species [77,525]. Evidently, in humans, as heterotrophic organisms, the forms in which we can use energy are limited in type, amount, and proportion, to those pre-formed by our preys in their constitutive structures (i.e., the food we eat). Our homeostatic systems need to dose/limit the use of substrates for energy and establish dynamic reserves with enough to survive even under severe availability problems, but not as much as to limit our mobility and performance with the reserves’ dead weight [526]. In this sense, humans are not, by far, the most evolved omnivores, but historical ingenuity and resilience have developed a working blueprint (Neolithic) that we maintain nowadays with little change. The periodic intake of food follows environmental rhythms that add to our internal ones [527,528,529,530]. Specialized tasks, such as group hunting, food gathering, and, when possible, (limited) preservation of food, help maintain a relatively steady supply of nutrients for the group. The energy density of the human diet is fairly high compared with strict herbivores, and is comparable even to that of carnivore predators. The dual ingestion of plant and animal foodstuffs allows for a wider variety of edible sources [531], and has established some sort of periodicity in their availability and spurned the rearing of selected plants and captive animals to extend our margin of food security [8].

Figure 3 shows a crude representation of the theoretical components and points of variation in the energy balance of humans. Practically all usable energy (*energy intake*) comes from the food ingested, a part of which is lost as waste (and to fuel commensal–symbiotic biota oxidation): *waste energy*. The (‘net’) nutrients supplied by the diet are processed and brought, mainly, to the liver, and are subjected to a rapid triage and partial catabolism; this energy is further filtered/classified and used specifically (in the process of energy partition) for both energy and to support the body’s components’ turnover (and growth/biological contribution, if any). A large part is used to fuel our physiological functions, but unused energy is (primarily) accrued, with all the caveats referent to possible unwanted negative effects [532,533], or is oxidized for heat production (i.e., *inefficiently* from the metabolic point of view) when above the homeostatic needs [534,535]:
**E-intake** *crude: food* − **E-excreted**
*heat and waste* − **[E-transfer]** = **E-net-balance** *loss/accrual*

This is the simplified standard energy balance equation with the important difference, with respect to the classical equations, of taking into account the energy transfer, a factor that establishes quantitatively significant time- and sex-related differences between individuals. Its importance is quantitatively considerable and, essentially, corresponds to the generation of new individuals of the species, including, eventually, their sustenance during neonatal life (lactation) or even further. The direct energy transfer contribution of females is enormously greater than the testimonial one of males, which, in addition, is massively lost, unused, as waste.
Figure 3**Energy balance.** In black, explanation of the component; in red, expression of the component as part of the global energy balance equation. The components of the energy balance listed correspond to the management of energy taken from the medium and that returned to it, including the net gain/loss of energy accrued, but introduce an ‘extracorporeal’ element which we can define as biological contribution to the social group.
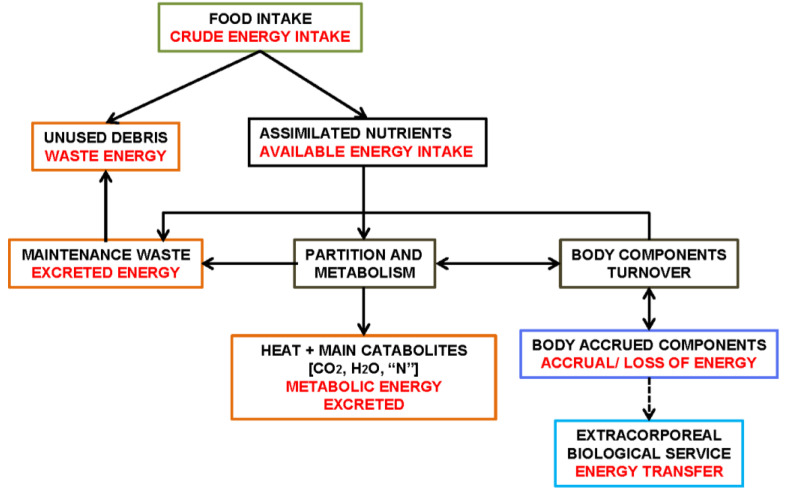



### 4.2. Dietary Energy Assimilation, Triage, and Partition

Foods are a complex mixture of (often alive when eaten) cells and tissues, plus other biological structures and materials, partly treated by heat or biological agents (i.e., cooked, fermented, pre-digested, dried, diluted, etc.) with a considerable degree of variation: easiness of ingestion and digestion, abundance/sufficient overall mass, and nutrient composition. All these materials are processed by our digestive tract (and microbiota) to produce a wide number of simpler chemical compounds, belonging to a short list of types of molecule (i.e., fractions) that facilitate their utilization and eventual disposal. There is also a residual fraction of indigestible or insufficiently processed waste, which is discarded as feces. Most of the small molecules formed in the breakup of nutrient macromolecules, followed by other common biochemical processes carried out by our digestive enzymes (and the additional action of gut microbiota), are released at varying rates (and times) within the intestinal lumen. The result of the (partial) stomach digestion of food (chyme) is further subjected to a longer and more thorough intestinal digestion. The hydrophilic components are absorbed by the intestine wall cells and transferred to the blood, being carried to the liver via the portal vein. The most lipophylic compounds, however, are carried through the intestine walls into the lactaries of the intestinal lymphatic system, forming the chyle. This lipid-laden lymph is eventually (and slowly) poured into the cava vein, close to the heart, to obtain a maximal dilution of its (potentially thrombogenic) load.

These variable mixtures require a further refining/protective process of triage and partition. The heavier part of the process is carried out by the gut, producing small-size molecules from complex food mixtures. Figure 4 shows the (main) changes in the qualitative presence of metabolites and other compounds successively over time in the intestinal lumen, portal blood, liver and, finally, the (cava vein) systemic blood, from which most other cells/tissues draw their needed nutrients, thus presented in (homeostatic) canonical molecular species and concentrations.

The complexity of the mixtures of compounds present in the intestinal lumen is considerable (and fairly variable, depending on the last meal composition). The first triage step takes place in the intestine, where most lipophilic compounds are filtered into the lymph to form the chyle. The hydrophilic small molecules cross the villi blood capillary bed walls to the blood carried by the porta vein to the liver in a complex mixture filtered only by solubility and small molecular size. This continuous process represents the first biophysical separation of compounds depending on their charge, hydrophobicity, and size. The larger macromolecules remain in the process of digestion within the intestinal lumen until the digestive/extractive process is exhausted, i.e., most extractable small molecules have been absorbed, and the indigestible remains continue in the gut until they are further digested in the colon and later excreted. Nevertheless, this process is relatively rapid and efficient, simplifying the next step: nutrient-derivatives partition by the liver, essentially. The products taken from intestinal digestion via porta blood and brought to the liver have three main paths to be processed by: Passing through the liver and leaving through the supra-hepatic vein blood, eventually voiding into the lower cava vein. This efflux carries amino acids, glucose [536,537], lactate [538], ketone bodies, bicarbonate [539], and lipids, such as lipoproteins [540].Excretion/waste through the bile to the gut. The bile is a partly toxic mixture but also contains bile salts that act as emulsifying detergent [541].Implication of xenobiotic inactivation mechanisms [542,543]. These processes are complemented by the extra-hepatic dumping of chyle in the cava vein.Metabolic transformation, in which the input molecules are modified by the liver cells and finally driven to one of two main functional fates [133]:4.1Oxidation: to obtain energy, e.g., conversion of substrates to 2C and oxidation to CO_2_ through the Krebs cycle.4.2Storage: to save energy in the form of depot substrates (TAGs, glycogen) stored in the liver or exported (point 1).

The incidence of the overall dietary energy of microbe-generated catabolites, including short-chain fatty acids such as acetate [544], propionate [545] or butyrate [546,547], is not commonly taken into account, at least with respect to their eventual quantitative importance as sources of energy and other possible effects on energy balance. The disposal mechanisms of ingested ethanol [548,549], including those generated by the gut microbiota [546,547], and the even more toxic methanol [550], are well known, as are their metabolic/health consequences. In contrast, the direct intake (and the added microbiota production) of a number of compounds in food uncommon to our biochemistry should not be counted as nutrients. This is the case of D-lactate, often considered a regular nutrient, in spite of its slow oxidation rate and the fact that most of it is excreted, unused [551].

### 4.3. The Importance of Protein as Energy-Providing Nutrient in Humans

Protein is a critical nutrient for all animals because it is practically the sole source of both amino N and the hydrocarbon skeletons of essential amino acids [506,552] (as well as of sulfur) [553] for non-autotrophic organisms unable to synthesize them. They are needed for the synthesis of protein and a large number of essential N- and S-containing compounds needed to sustain and transfer life. The role of essential amino acids is critical for heterotrophic organisms deficient in pathways to synthetize hydrocarbon structures, as is the case for humans [554]. Thus, in many animal species, they are generally spared (as much as possible) from oxidation as energy substrates. This line of thought has maintained a peculiar niche for dietary protein in the mind of many metabolism-oriented scientists, who have generally relied upon the biological trend of preserving protein as a critical provider of anabolic substrates rather than (also) seeing it as another source of raw energy, as is the case for most carbohydrates and fats [520,531]. This is true qualitatively, but the relatively large amounts of protein ingested by some omnivores such as humans are (quantitatively) well over the minimum amino acid supply required to maintain N turnover (i.e., such as those of protein and nucleic acids) [555]. The biological maintenance of this extra supply of amino acids means that they are too a prime energy component of our diet [556], in a range that is lower than that of carbohydrates but higher than that of lipids in many human societies.

The ‘excessive’ consumption of protein provokes a direct additional biochemical problem: that of how to dispose of excess amino N [557], which compounds the effects of the indirect (albeit common) association between the joint consumption of high protein and fat [558]. In any case, humans have consumed protein in large amounts, when available, from very early in their evolutionary process [7], a trend shared with the great apes (most of which, however, remain largely omnivore/vegan but meat-supplemented) [559]. Even in those humans following strictly vegan diets, however, the relatively lower presence of protein in the diet [560] does not result (apparently) in crippling or generalized deficits [561]. Omnivorism provides a fairly higher dietary supply of protein to cover more than the strict needs of amino acids. Most problems with dietary protein are a consequence of the quality of protein (i.e., abundance and availability of critical amino acids, digestibility) rather than bulk.

The advantages, for eclectic omnivores, of leaning towards carnivorism (in fact, towards a higher availability of apparently unneeded extra protein in the diet) have been recognized as a way to facilitate growth, protein deposition, and better energy use [562,563]. Increases in protein with respect to carbohydrate intake induce, in rodents, a decrease in body fat [564,565] and better control of glucose [566,567]; these effects are also observed in humans [568,569], with some beneficial effects on MS [570]. In rats, an increased intake of carbohydrates, coupled with lower protein, decreases lipolysis and enhances fat deposition [571,572]. However, an adequate provision of carbohydrates may stimulate growth and predator efficiency [573]. It seems evident that the energy provided by carbohydrates may be best used with the adequate presence of protein in the diet. This is well known via human lore, but has not been systematically analyzed, probably because of the extreme difficulties of obtaining an adequate model of study.

There is little doubt that protein is a necessary component of the human (and that of every other animal) diet, in excess of its known provision of amino acids for the synthesis/turnover of protein and other N-containing compounds. The question here lies in the quantification of these needs. Diets very poor in protein are shunted by some omnivores, [574] including humans, and the absence or low level of some amino acids (or simply of enough amino N) spurns malnutrition [503,506], which may evolve into serious neurodegeneration [575]. Milder insufficiency of dietary protein may also accelerate the development of a number of diseases, such as obesity, within the MS spectrum [576,577]. The main effects observed include limited growth [578] and reproduction [579,580], negative N balance, and sarcopenia [581]. The clear need for amino acids in the diet, further than the essential amino acids and amino N provision, cannot be justified solely by these aspects, even when including the need for amino N to construe the large amount of essential molecules that incorporate this element, such as nucleic acid bases, porphyrins, or even small signaling molecules [506]. Notwithstanding, and despite their essential functional importance, the mass proportion of dietary N incorporated into these non-protein biological N compounds is relatively small [582]. It is even easily covered by the N-poor leafy diets of many herbivores [583,584], leaving a margin to spare for biological transfer during reproduction. The partly ‘granivore’ human diet contains (obviously) a significant amount of plant protein, in addition to 6C (6-carbon carbohydrate, i.e., oses) providers and some fats. The primarily 6C- and 2C-producing nutrients are processed more efficiently by the omnivores adapted to complementary protein in their diets [133], such as humans and rodents. Protein is also directly used for energy along with these hydrocarbon substrates in a complex multidimensional selection of nutrients favoring both growth and energy [585]. The provision of N by most of these ‘rapidly oxidized’ amino acids is just excreted, unused [196], which leaves the peculiarities of the wide array of hydrocarbon skeletons of some amino acids as a real, energetically significant nutrient, of which global importance is seldom recognized as such, but perfectly identifiable in the diet of humans and other omnivores.

The common consumption (often in excess) of carbohydrates and fats, including many industrially processed products that render ‘purified’ foods proportionally enriched in sugars, starches, and fats, may result in deficient diets. Often, the ‘original’ protein fractions (i.e., casein, whey, gluten, soy protein, etc.) are used for industrial purposes (including other elaborated food products). These diet manipulations may affect the effects of the diet by loss or alteration of their initial biological food composition and structure [586,587]. The obvious (albeit not based-on-knowledge) general opinion is that the supply of sufficient protein to cover the N and essential amino acid needs (for turnover and accrual) is sufficient for normal function, i.e., leaving the supply of raw energy largely (but almost completely) to starches and fats. The continuous use of this factual ‘excess’ dietary protein consumption, and its maintenance for health during growth, pregnancy-lactation, old age, and disease, has been tactically acknowledged via facts, but there is of yet only sparse literature recognizing and establishing this too-common human feeding trend. Unfortunately, misguided observations have been used to promote the assumedly ‘healthy’ intake of low-carbohydrate ketogenic diets (especially those with reduced energy content) [588,589]. Ketogenesis (in adults, since maternal milk constitutes an example of growth-promoting obesogenic ketogenic diet [590] until weaning) is an emergency mechanism to distribute pre-metabolized 2C providers (i.e., 3-hydroxy-butyrate or acetoacetate) between most tissues (including muscle) to provide rapidly usable energy [591,592] and prevent/limit the forced storage of 2C as TAG, given the low-temporal capacity of the liver mitochondria to oxidize any massive flow of 2C consequence as of excessive dietary carbohydrate (6C hexoses glycolyzed to 3C and, eventually, yielding in 2C as main final energy staple) [593,594]. The so-called ‘refined diets’ and the excessive use of sugars and TAG in our daily food intake enhance the need to oxidize the massive availability of 2C in order to remove the excess energy *already* in the system [595,596]. The (emergency) alternative is the derivation of 2C to lipogenesis and the deposition/dumping of body fat [597,598]. The ‘solution’ adopted when using ketogenic diets is to reduce the breakdown of 6C to 2C by lowering 6C but promoting an even higher excess of 2C via breakup of dietary lipids. In this case, the excess of 2C cannot be easily used for fatty acid synthesis, since its β-oxidation is already the main source of 2C. Thus, the excess carbon is driven towards ketogenesis and the export of ketone bodies as a rapid energy supply for most tissues [599,600]. The main problem (excess of 2C supply with respect to energy needs) remains, and is simply ‘exported’ to peripheral tissues, often not fully designed to use them as main energy substrates. In addition, ketone bodies induce metabolic irritation, causing acidosis [601] and, eventually, serious bidirectional alterations in the control of insulin and glycaemia [602,603,604]. Dietary protein may in part limit these problems by providing hydrocarbon skeletons that partly compensate (see below) the problems elicited by lipid-laden ketogenic diets.

There is little information on how much “extra” protein is needed to help metabolize and dispose of eventual excesses of other energy substrates, primarily, carbohydrates [555,605,606,607]. A direct relationship has not been yet established in practice, despite the fact that the biochemistry of the whole process has been known for many decades. One of the critical factors affecting interpretations of the need for a regular ‘extra’ consumption of protein in human diets is, precisely, the ample variety of pathways and molecules (albeit not so many end-products) implied in their catabolism, which may limit their canonical analysis. Often, unfocused powerful metabolomic tools are applied to ‘simplified’ experimental set ups, usually with incomplete results and partial interpretations. The adscription of this additional role of dietary protein in the anaplerotic activation of the Krebs cycle [608] fits with the postulated role and presents a complementary view to the large amount of information already available on the regulation of mitochondrial oxidation of energy substrates. The anaplerotic process allows for a full function of the Krebs cycle by supplying sufficient amounts of the intermediate metabolites needed to process the raised flow of 2C units to yield energy. These compounds are largely regenerated in the cycle operation, but their higher concentrations [609] increase the overall flow of carbon (2C → CO_2_) manifold.

A relatively large group of intermediate amino acid catabolism compounds (hydrocarbon structures, after losing their amino or sulfur moiety) are eventually processed to become intermediate compounds of (mainly) the Krebs cycle itself. These hydrocarbon skeletons, largely used in their synthesis (by plants, and other organisms, but not animals) were initially part of this cycle, as are the catabolites produced in their metabolism. This particular group of small molecules, formed (mainly) along the catabolism of dietary amino acids, can be roughly defined as a group because of their purported function as KCAIs (Krebs cycle anaplerotic intermediates), since they induce an important anaplerotic effect on the flow of energy through the cycle, increasing the rates of oxidation of 2C units (and of 3C, obviously, as a main source of acetyl-CoA) with a concomitant increase in NADH and ATP generation. The KCAI constitute not only a majority of amino acid catabolite structures (i.e., producing 2-ketoglutarate, succinyl-CoA, fumarate, and oxaloacetate), but also include the propionate produced by the oxidation of odd- and some methyl-branched chain-fatty acids [610], as well as being produced by the intestinal microbiota from a number of dietary components [611]. This anaplerotic effect has been widely studied in relation to the needed higher availability of oxaloacetate/malate, mainly from pyruvate, precisely to potentiate a higher flow of 2C oxidation through the Krebs cycle [612,613]. The direct relation to dietary amino acids as a regular (needed) source of KCAI further exploits these seminal studies to relate the production of additional oxaloacetate via insulin activation, not only driving to gluconeogenesis [614] but to the direct common function of the Krebs cycle in its main role as oxidizing/powerhouse machine of the cell mitochondria.

Figure 5 shows the paths for the incorporation of metabolites resulting from the catabolism of aspartate, asparagine, arginine, glutamine, glutamate, proline, tyrosine, phenylalanine, valine, isoleucine, methionine, threonine, histidine, and tryptophan. The global proportion of these KCAI-providing amino acids in relation to total amino N dietary handling is in the rough range of about half (or more) of the moles of amino acids ingested as dietary protein by humans; this group also accounts for a comparable (or even higher) proportion of those oxidized, with their hydrocarbon skeletons being converted into 2C, 3C, and, especially, KCAI (which, eventually, revert to 3C → 2C → CO_2_). It is important to add that the amino acids yielding pyruvate (alanine, cysteine, serine, threonine, and hydroxyproline) may, eventually, provide further KCAI via the carboxylation of pyruvate [615,616]. However, the pro-gluconeogenic (i.e., low 6C availability vs. high 3C or 2C) conditions needed for an enhanced supply of oxaloacetate through this path are quite different from those of the postprandial state in which the 2C catabolic effects of KCAIs have been described [133]. The KCAI synergistically facilitate the effect of insulin on carbohydrate catabolism and oxidation of 2C by increasing the flow of carbon through the Krebs cycle, consequently limiting the negative effects of the increased alternative outlets of 2C: ketone bodies, fatty acids, and TAG, which interfere with the energy metabolism by eventually depositing surplus 2C (as TAG) in practically any available space of the body [617]. This known effect and inescapable fate of amino acid catabolites may help explain the biological and functional advantage of *using dietary protein for energy* in the context of an abundant supply of 6C (or 2C). In fact, all canonic dietary protein amino acids (including the imino acid proline) can provide KCAI in their catabolism, with the sole exception of glycine (primarily), leucine, and lysine.
Figure 5**The ‘*Krebs Cycle Anaplerotic Intermediates*’ (KCAI).** Most KCAI are generated by normal amino acid catabolism in human tissues. Other KCAI, such as propionate, are produced by the microbiota or as catabolites of odd-number carbons or methyl-branched fatty acids through β-oxidation. Krebs cycle: pale green rectangles and black solid arrows. Other intermediate metabolism: white rectangles, purple arrows. Lipid metabolism: yellow squares and purple arrows (AcAc = acetoacetate; FA = fatty acids). Amino acid catabolism: blue rectangles and blue arrows (alternative paths are marked with dashed lines). 1C represents the one-carbon pool metabolism. The amino acids susceptible to directly generate KCAI are marked with a green star (Hyp = hydroxyproline). In all, two/three-thirds of amino acid structures can generate KCAI. Those producing pyruvate, however, may eventually provide KCAI via pyruvate carboxylation.
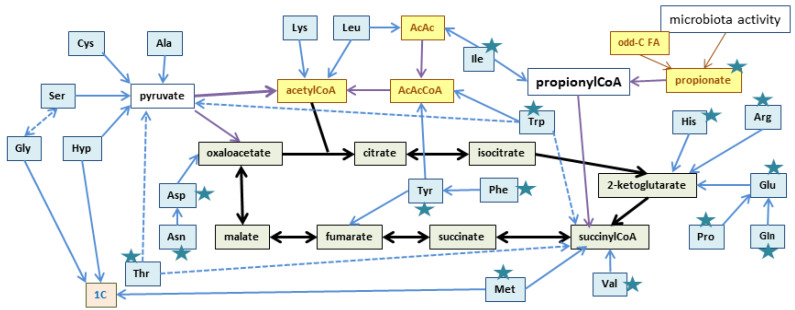



### 4.4. Influence of the Timing of the Digestive/Absorptive Process on Nutrient Energy Partition

The absorption of a few small non-charged molecules (i.e., ethanol) is rapid and may take effect shortly after ingestion through the stomach walls [618]. The stomach easily voids its contents to the duodenum when containing only water or carbohydrate solutions [619], a process affected by many factors, including dilation or exercise [620]; however, voiding is slowed by the presence of fats (in part to match their entry into the intestine with the release of bile to emulsify them) and/or protein [621], for which digestion already starts in the stomach by the pepsin gastric-juice-acidified medium. Under these conditions, the stomach releases, periodically over a varying timespan, small bursts of partly digested gastric content, which are neutralized in the small intestine and exposed to the bile as well as intestinal and pancreatic secretions. The absorption of free sugars (i.e., glucose) follows the pace of stomach voiding [622], and takes place in the upper part of the small intestine [623] at rates close to those of hydrolytic release. In the duodenum, the emulsified lipids also begin to be hydrolyzed to yield free fatty acids and start [624], under non-acidic conditions [625], the long process of completing protein denaturation and hydrolysis. These processes have different speeds depending on the nature of the reactions undertaken, the concentrations, and the upregulation of hydrolytic processes via removal (in this case, mainly absorption) of the reaction products (such as oses, fatty acids, and amino acids). Digestion also shows different topical relations within the gut, with the hydrolysis of most of the main nutrients and their absorption taking place in the small intestine (mainly in the first segments), both in the wall (and annex structures) and lumen in a complex integrated system [626], leaving the colon with the task of recovering water (and minerals), extracting the last usable molecules, and the structuring of waste into fecal excreta. Figure 6 shows a schematic approximation of these different rhythms of release of nutrients from the chyme to either the porta system (and later to the liver) or the lymph system. The human stomach, usually, empties in a few hours or less (even minutes, largely depending on the mass and type of the components) [623]. The small intestine transit is covered in a few hours, again with considerable variability, and the stay of the residual intestinal content in the distal part of the intestine may take (additionally) from a few hours to more than one day. In any case, most of the 6C from monosaccharides or hydrolyzed oligosaccharides is absorbed at a rate close to that of stomach voiding [622,623]. More 6C (including pentoses, eventually recyclable to 3C/6C) and 3C fragments derived from easily-hydrolysable polysaccharides follow suit. Thus, the main load of carbohydrates may arrive to the liver between minutes and a few hours after a meal. The fats take more time, first because of the added difficulty of handling lipids in the hydrophilic intestinal medium and the conditions needed for lipase activity [621,627], as well as due to the slower transfer to the lymph [628] and its longer journey up to the thorax and into the blood [624,628]. In sum, the bulk of fat-derived nutrients (except free glycerol, a 3C, and a few other small molecules) [629] are carried to the systemic blood later than the first wave of carbohydrate [624]. The complexity of proteins’ structure deeply affects their digestibility [630]. A significant part of dietary protein may end up in the feces because of incomplete hydrolysis, resilient structures of the food tissues, lipid-protein interferences, etc., but also due to variable microbial activity, which helps extract part of their amino acids or metabolites, mainly in the colon [631]. The inescapable need for three main steps in protein hydrolysis: (1) denaturation to make the peptide chains accessible to enzymatic hydrolysis, (2) protease hacking of the polypeptide chains into peptides, and (3) peptidases breaking up the peptides into amino acids, requires a time-consuming sequence of breaking-up (and absorption) that is longer and more complex than that of TAG and starches; consequently, the protein catabolites are transferred (mainly) to the portal vein at quite variable rates [632,633]. The digestion process becomes more complex by the additional action of enterocyte cytosol/membrane peptidases [634,635].

The longer stay of protein (and complex carbohydrates) in the gut also results in a higher rate of usage by the symbiotic and commensal microbiota, of which growth, energy wasting, and accrued energy (most of it lost in the feces with their bodies) reduce the availability of part of our dietary energy, but also make available part of the energy remaining in indigestible material (fiber, including that carried by the foods and some proteins and lipid–protein complexes, limit dextrins, etc.) that the biota may hack for their (and our) own utilization [631]. 

The time difference in the arrival of the bulk of monosaccharides and that of amino acids to the liver aggravates the problem of a transient overload of 6C arrival, as well as the concomitant need for KCAI (untimely not fulfilled because of the delay in amino acids arrival to the liver) to help efficiently dispose of the excess of carbohydrate-derivable energy. The disposal of 6C to first 3C and then 2C poses no unsurmountable problems, but the management of 2C does. The 6C load may be transiently controlled by letting glycaemia rise and (to a much lower extent) topping the liver glycogen stores; nevertheless, the obvious ways are the glycolytic production of 3C, which may be more easily exported and, definitively oxidized to 2C. The latter can be either oxidized through the Krebs cycle or used to synthesize fatty acids (and, in this case to a limited extent, ketone bodies). The excess of 2C is, again, a serious problem to solve: the accumulation of fatty acids as TAG (and where are they dumped) could be an emergency (temporal) solution but may backfire because it is unnecessary and logistically complicated. The only effective way to solve the transient excess of energy is to burn it, preferably in the canonical way, as 2C in the liver by increasing the flow of carbon though the Krebs cycle with the expected help of KCAI. The conundrum of handling a ‘time gap’ between the surge of carbohydrate-derived 6C → 3C → 2C and that of protein-derived KCAI is difficult to handle, since the immediate way-out of the emergency is the storage of TAG instead of the oxidation of 2C. The core problem may not even be related to dietary lipids, but to an excess of carbohydrates. The persistence of this problem over time may, probably, cause a condition of (a) hepatic steatosis and (b) hyperlipidemia, complicated with (c) ectopic dumping of fat elsewhere.
Figure 6**Nutrient-related differences in the time needed for the substrates released through digestion to reach the portal or systemic blood**. This graph has not been generated using hard homogeneous published data, but only textbook and scattered information on the time taken for digestion of the dietary nutrients derived from (an extremely wide range of) diets containing carbohydrates (largely sugars, starches, and resistant starches), proteins (of relatively high digestibility, low, or resistant), and fats. Ethanol (exogenous) is a 2C included as comparison. The time periods presented are lax and imprecise, but include the stay within the stomach and gut, as well as the time taken for lymph to be transported along the large thoracic conduct to the cava vein. The time taken for blood-carried substrates to travel from the intestinal wall to the liver has not been considered, given the extreme lassitude of the time marks used compared with the shortness of the cava and its high blood flow. The last two lines include an appreciation of the large amount of fluid moved along the process of digestion between the complete digestive tract and the rest of the body: first, facilitating an adequate milieu for blending, hydration, digestion, dissolution, absorption/triage, and deposition within the gut; second, the reabsorption of water (and electrolytes), together with nutrients and (soluble) waste molecules, with the final dehydration and conditioning of feces for smooth expulsion. The lines at the bottom of the graph show the main (arbitrarily established) period of absorption of carbohydrates, protein catabolites (mainly amino acids), and products of fat digestion. Despite all the caveats and extreme variability in the processes studied, a gap exists between the peak of carbohydrate-derived nutrients and that of protein-derived nutrients, which may be higher for dissociated diets (even when this dissociation does not refer to days but to meals).
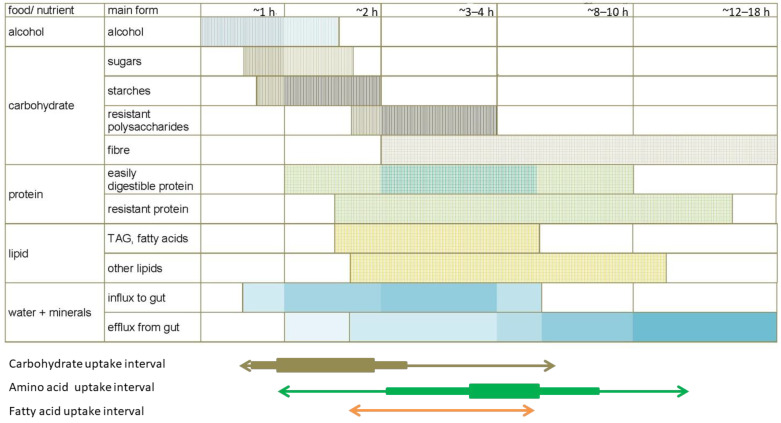



The few ways and means to shorten the interval between the processing of carbohydrates and proteins in the diet are largely known and available (albeit only potentially/partially effective), and can be summarized in the following:(a)The distribution of the daily food allotment should be spaced over time to allow for smaller portion sizes of food in each meal as a way to facilitate streamlined digestive processes (and thus their timing) and to prevent substrate overloads. Large portions and a higher frequency of meals both increase the net energy balance [636], and large servings also increase insulin resistance [637], but the effect on total energy intake in a higher number of meals is more limited on these parameters [638].(b)All meals should contain a varied mix of nutrients for optimal assimilation, but (in humans) the ingestion of carbohydrates (in special digestible starches and sugars taken in large amounts) should be combined with the presence of protein. This protein component may provide the KCAI and other small molecules that prevent the almost exclusive and massive availability of 6C (eventually building an excess of 2C). Meals with a large carbohydrate and protein presence actually facilitate adipose tissue lipolysis [639]. Diets laden with carbohydrates and lipids, on the other hand, distort the maintenance of glycaemia [640]. The differential increases in protein, globally, do not alter energy intake [641].(c)The proteins referred to in point (b) should contain a reasonable diversity of molecular species (translatable to quite different velocities of hydrolysis) to facilitate the distribution, over time, of their digestive breakup products. Easily digestible proteins such as whey (as compared to the slower digestion of casein) [632] may help fast-start the protein effect on carbohydrate digestion. The strict N protein needs are easily covered with equilibrated omnivorous diets, and the body total protein mass is maintained despite important variations in the amount of dietary protein ingested [642]. However, dietary deficiency in protein may affect growth and body protein when not corrected [579], which hints at the primary importance of protein as a key factor in facilitating the energy use of dietary carbohydrates, since the preservation of dietary protein amino acids (as newly synthesized body proteins) to maintain body N homeostasis may be also affected by persistent dietary amino acid deficits.(d)Meals should contain enough protein (and/or lipid) to pace the stomach voiding in order to (at least) prevent fast and steep rises in circulating 6C, as explained above. The presence of fiber (instead of highly hydrolysable purified starches or oligosaccharides) may also help slow the overall process of digestion and, thus, allow for a smaller gap between the blood peak of sugars and that of amino acids.

### 4.5. Main Constrictions and Rules Governing the Partition, Use, and Disposal of Nutrients

The triage of the mix of nutrient-related molecules arriving to the splanchnic bed, mainly to the liver, in a first-pass from the porta vein, is a critical step in the efficient metabolic use of nutrients extracted from the food ingested (and digested). This process needs to be rapid and efficient in order to prevent the eventual complications of accumulation (metabolic, interactive, and even osmotic effects) or excessive dilution/saturation of the mechanisms that process them. The paucity of absorption, and the method of access of nutrients to our internal core homeostatic system, facilitate (being probably designed and evolved to carry on) the sequential handling of dietary nutrients. The most critical point of this process lies, perhaps, on the hepatic sorting, in a first-pass, of most of the nutrients assimilated. Partition has to be executed in synchronicity with the waves of nutrients carried to the liver by a high portal blood flow (to allow for tolerable nutrient concentrations in portal blood) [643]. The crude mixture of nutrient molecules brought from the gut has to be ‘adapted’ to our homeostatic requirements for overall distribution and use, a postprandial role for liver helped by the other organs of the splanchnic bed. Substrate partition, however, needs to leave sufficient slack for the implication of additional complementary processes in order to limit the effect of discordances between what is being received and the actual processing capabilities of the system in real time [132]. In fact, this complex task is made easier by a few main metabolic conditions ruling the overall process, which is evidently constrained also by the chemical nature and marked diversity of the nutrients to be processed for energy and maintenance.

(a)Glucose is a highly reactive molecule that spontaneously glycosylates proteins and other compounds [644]; in addition, its two main isomeric forms (in biological fluids) are maintained in equilibrium, but only the β-pyranose form is actively used by most metabolic pathways, with the α-isomer being slowly reconverted, thus slowing (and diluting) glucose utilization [645]. Glucose is, by far, not an ideal substrate, but it is the best one (and the *only* one) available. Eons of evolution of photosynthesis have produced this particular compound, which is now the main energy staple of plant metabolism. Simply, there is no alternative for heterotrophic organisms, such as humans. Thus, we have had to rely on plant-synthetized glucose as our main energy staple despite its many inconveniences. Consequently all animals’ energy homeostasis is centered on glucose [150].(b)Glycogen is the main glucose (6C) form of storage in animals and is related to plant starches. It is found in limited amounts in mammal tissues; its maximal levels (mainly in liver and skeletal muscle) amount to a tiny proportion of all body energy reserves. Glucose availability from glycogen is relatively fast (but not sustainable over time) [646] and, thus, amino acids (and other 3C sources) may be used for gluconeogenesis to cover on-spot needs or starvation-related deficits in glucose [647,648].(c)The main energy reserves of mammals, including humans, are TAGs and body protein. In fact, protein is not a canonical reserve as such, but its utilization allows for a margin of consumption linked to normal turnover. This arrangement includes a large part of body protein as a variable unit of energy (and amino-N) and dynamic reserve. Lipids are the best option for energy storage in animals because of their high energy-packing capability in an unbeatably smaller added weight, several-fold lower than that of glycogen and protein (i.e., hydrated vs. fully lipophilic, and energy-dense TAG structures); thus, lipids are the energy reserve substrate of choice [649].(d)The availability for use of the energy of lipid reserves is slower than that of glycogen, but both are coordinated [650]. To use the 2C pool of our fat reserves requires a progressive activation system compared with the direct glycolytic consumption of dietary 6C for energy. Despite the complex timing, application of catabolic protocols, and partial sparing of nutrients, at the end of triage, all nutrient sources marked for oxidation to obtain the energy we need to function are converted into 3C or 2C, but, eventually, to 2C, which is oxidized to CO_2_ in the mitochondria to provide energy [133].(e)Under conditions of energy deficit, the body’s reserves, especially protein [651], tend to be spared by decreasing energy consumption, leaving their use for energy metabolism mainly to cope with “non-consuetudinary situations” but common situations, such as starvation [608,652] or increased energy needs; energy transfer (reproduction) [653,654]; travel and migration (i.e., of birds) [655,656]; or illness [657,658]. The level of reserves should maintain a balance between possible time-extended needs and the mass of fat weight, i.e., the cost of storing and moving around the fat reserves as a hindrance to fight or flee, both as predator or prey, as well as migration and survival during the sparse-food-availability periods [656,658]. Protein “reserves”, such as those of plant seeds, are not present in humans, as the term ‘reserve’ refers to the whole-body protein content, maintained through a constant turnover from amino acids and energy. Deficits in dietary N sources ease the use of amino acids for energy instead of protein turnover, diminishing whole-body protein in a multi-fractioned way. Excesses of energy and amino acids facilitate the accrual/growth of protein, depending on a strict genetic control.(f)All systems are conditioned to save/spare and defend some strategic nutrients from wasting, especially those more difficult to obtain and retain, to maintain working pools and to prevent deficits. The list includes minerals, such as P, Ca, Mg, Zn, Fe, I [659,660,661], and ‘essential’ structures, such as some amino acids, most vitamins [662,663], and polyunsaturated fatty acids [664].(g)The preferred hierarchy of use of external (diet) substrates for energy depends on their availability, the need for energy, the metabolic context, ability and predisposition to use them [665], and the immediacy of their presence already in the gut. The first staple to prioritize for energy production should be the dietary fatty acids (PUFAs and others are spared, and in fact picked up, to a point), oxidized to obtain energy (via 2C). The second, depending on the overall availability, are the carbohydrates (6C and their derived 3C); meanwhile, protein is, assumedly, the last to be massively consumed. In fact, a whole-body energy status is reached in which the flows of substrates are mixed and adjusted to an optimal efficiency and preservation of both energy and selected components [666,667]. Nevertheless, these considerations are essentially theoretical and, in practice, a sizeable part of dietary fatty acids may be saved, favoring TAG deposition [668,669], or protein may be oxidized to facilitate carbohydrate disposal, as indicated in Section 4.4.(h)Under the conditions of starvation and low dietary energy availability, the body applies a protocol of energy saving, lowering exercise, reproduction, thermogenesis, and maintaining activity and function barely over the survival level until (if) external conditions improve [520,665]. If no 6C are available, then TAGs (from food and/or reserves) are the substrate of choice [670]. In the last stages of starvation, body protein is more actively used [670,671], albeit in higher proportions when there is no concomitant presence of body fat [672]. The diet and functional orientation of different species deeply affect their management and homeostatic resilience to starvation [673].(i)Under conditions of high food availability (i.e., 6C, TAG, and protein available in excess of assumed N needs in every meal), the order of use in humans shifts to using excess 6C first, either to oxidize it or to eventually store part of its energy as TAG. Nevertheless, it has been known for long time that gross excess of protein in a meal (i.e., in the absence of large loads of carbohydrates) results in the selective rapid oxidation of the proteinaceous overload (i.e., a shift to ‘carnivore mode’ over the basic ‘omnivore’). The general indications of metabolic priorities must be taken only as general indications, since it is the complete metabolic panorama that in fact drives the mobilization of supplements [674]. Surplus dietary protein is used for energy, essentially, to prevent the accumulation of amino acids [557] (i.e., disposing of unwanted excess N), which results in the need to also oxidize their hydrocarbon structures. In part, this process accelerates the oxidation of 2C, but also helps create an unwanted overload of 3C (and, thus, potentially of 2C again), which may compete/interfere with the process of disposing of dietary 6C [675].(j)Dietary TAG, the main canonical source of 2C, may be spared in part because their availability for oxidation may interfere with the disposal of excess 6C, with both processes fueling the mitochondrial oxidative machine with 2C. The sidetracking of β-fatty acid oxidation is the lesser of two evils since its ‘preventive or temporal’ incorporation into the whole body TAG pool may give time for the maintenance of the postprandial energy equilibrium, [669,676] despite being a harbinger of delayed severe metabolic consequences [677]. But, a rise in circulating 6C and insulin resistance [678,679] are more urgent problems to solve than the eventual increase in fat stores [680].

These general postulates or ‘rules’ help keep us alive within an ample set of possible conditions of dietary energy availability. However, the emphasis (as marked in our genome) is secularly established on survival under scarcity (the most common situation encountered during all of our previous evolution), evidently not finding on ways to navigate out of a sea of available energy. The studies conducted on rodents, subjected to abnormally high-energy diets while the limited and further debilitated mechanisms that maintain energy balance were also blocked, have provided a massive amount of information that shows the marked obesogenic effect of hyperenergetic diets laden with TAG and 6C (especially with mono- and oligosaccharides) [681,682]. These diets usually contain other ‘perks’ for modern humans (uncommon in the wild), such as salt [683,684], in addition to increased palatability [685,686,687] and other behavioral factors, such as, simply, too many choices, satiety, and even boredom, the latter becoming more important when combined with sedentary attitudes [688,689,690].

Historically, we have gained, very recent, access to massively produced foods based on the intensive cultivation of grains and seeds, the also ‘industrialized’ rapid growth of animals for meat (such as fowl, pork, and veal), and the high availability of cheap dairy products and chicken eggs as important supplements of protein and fat. These achievements are apparently breaking the Malthusian predictions of scarcity, thanks to the industry success (irrespective of environmental preservation) of the ‘Green Revolution’ [691], which has finally allowed for the full-belly policies that have been sought for millennia. But, these amazing triumphs have a clear downward side: humans are not biologically prepared for this easy access to (historically) dream food availability. Our physiology reacts to abundance accordingly, to limit damages, but it is not enough, since pandemic-scale metabolic disorders are growing out of proportion. Nevertheless, and despite these problems, the food security levels already achieved have expanded the lifespan and quality of life of most of present-day humans.

In addition to these ‘rules’, the elegant simplicity of the process of partition (Figure 7) and diet assimilation that takes place daily in thousands of millions of humans attests to the extremely fine honing of these biochemical mechanisms over eons of evolution. These processes occur irrespective of the quite diverse conditions of age, health, ancestry, social and physical environment, diet, and other individual differences.

The development and maintenance of MS are highly influenced by the relative amount of energy ingested in relation to our real needs, and by the nutrient composition. When a problem lies in an excess of energy intake, the only way to face it essentially relies on: (a) trim or limit the flow of energy fuels ingested, and (b) facilitate/activate their oxidation: 6C → 3C, 3C → 2C (and 2C → CO_2_). This can be achieved only if, in addition to limiting food intake, these two premises are accomplished:A timing coordination between the arrival of portal blood substrates to the liver and their release (or that of their metabolites) to the systemic bloodstream. The rate of handling of these substrates by the liver should be sufficient to adapt the flows (in/out of the liver) to those of the transforming processes: 6C glycolysis to 3C; oxidation of 3C to 2C; oxidation of fatty acids to 2C; (eventually) lipogenesis from 2C; esterification of fatty acids to TAG; incorporation of TAG to VLDL; (eventually) synthesis of ketone bodies from 2C; oxidative catabolism of amino acids, starting by the disposal of their amino-N; and, then, oxidation to 3C, 2C, and KCAI. These transformations are coupled to the release of (mainly) glucose, VLDL-TAG, 3C (mainly lactate), amino acids, (eventually) ketone bodies, and waste catabolites. The liver retains (optimally) little in the form of net reserves, other than glycogen (if needed). The release is regulated to maintain the systemic blood levels of metabolites within homeostatic parameters.The liver fuels cover its needs for energy via oxidation of 3C to 2C and the oxidation of 2C through the Krebs cycle. The insulin-E2 regulative mechanisms must be sufficiently operative to allow for the sustained operation activity of the pyruvate dehydrogenase complex (3C → 2C) [184].E2 also favors the use of protein amino acids as energy substrates through the enhanced elimination of their N moiety. Consequently, dietary protein, irrespective of its constitutive N, must be catabolized, along the process of triage, to prevent the bottleneck of 2C accumulation (Section 4.4). Necessarily, this rapid use of amino acids requires an equally efficient system of amino N disposal, coordinated with the estrogen-insulin process of 3C oxidation, and circumventing the key rule of N preservation [152,196,692]. If all these premises are accomplished, then there should be a sufficient parallel supply of anaplerotic substrates (KCAI) to facilitate the rapid oxidation of 2C to CO_2_ and, obviously, more than sufficient biochemically available (NADH and ATP) to the liver, coupled, especially, with an efficient distribution of substrates to all organs and tissues.

It can be assumed that the specific oxidation of protein for energy is probably induced more for KCAI production and the availability of diverse small hydrocarbon catabolite structures, in general, than simply for energy alone (at least in non-strict carnivores), since the efficiency of energy extraction from protein is lower than from carbohydrates and fatty acids, at least down to the 2C level, with the additional costs of pathway diversity, as well as the need to circumvent the overlapping regulation of the critical processes of protein turnover and waste N excretion [693,694].

The canonic system of definitive amino N elimination is the operation of the urea cycle, but its strict control mechanisms are, in general, more oriented to prevent (or limit) the loss of (difficult to obtain) 2-amino-N, thus defending the retention of body amino acids. In this respect, T may also play an additional inhibitory role on nitrogen wasting [695]. The whole process is regulated by diet [696,697,698], glucocorticoids, and other hormones [699,700]. The first step, the production of ammonia from 2-amino groups, is widely distributed via glutamate dehydrogenase [701,702] (acting also in the synthetic direction), hydroxy-acid dehydratases [703], the glycine-cleavage system [704], the purine nucleotide cycle [705,706], diverse deiminases and deaminases [707], etc. But, the final pairing of ammonium and an amino group requires the presence of preformed urea cycle-intermediate amino acids, such as arginine (and/or ornithine/citrulline) [692,708,709], accounting too for the multiple needs of arginine (e.g., for the synthesis of nitric oxide [710], post-translational Arg-methylation, and deamination to citrullinyl residues [707,711]).

The existence of an alternative pathway producing nitrogen gas (N_2_) from 2-amino groups has been repeatedly postulated, and experimentally proven [133,712], but the actual pathway in mammals has not yet been elucidated [133]. The gross analyses of nitrogen balance, however, have excluded all tested explanations other than the release of N_2_ gas [712,713] in addition to canonic urea excretion [714]. The alternative N excretion path [715,716] has been related to arginine [692], and its activity is higher under conditions of excess protein N and energy availability [692,712,715]. The implication of E2 in the process has been also observed [196]. These data point to an eventually increased amino acid oxidation being part of a mechanism to help dispose of excess carbohydrate energy (Section 4.3). This mechanism, undefined as to its specific biochemistry but showing both metabolic consequences and regulation, may help explain not only the effects of nutrient overloads, but also those of some ‘refined’ diets (i.e., made up of only a few purified simple components, often with little attention to varied protein content) that favor inflammation and fat deposition [586]. The ability to oxidize carbohydrates easily and completely is hampered by grossly obesogenic diets (such as the cafeteria formulations for rodents [682,717]). The maintenance of an insufficient flow of 2C through the Krebs cycle is, probably, the main individual factor inducing hepatic steatosis (or NALFD) as a plausible first step of MS [171], a condition directly related to insulin resistance (type 2 diabetes) and obesity [718,719]. Later, other components of MS eventually continue developing their actions to eventually construe MS in its full pathologic splendor.

## 5. The Connective Tissue: Role in Energy Handling and Inflammation

### 5.1. Structure and Functions of the Connective Tissue

The complex, diverse, but somehow coordinated nature of the MS cluster of disorders clearly points to its causes, i.e., inflammation, not being the sole consequence of an organ-specific primeval disorder. The common direct implications of both obesity and insulin resistance (up to type 2 diabetes) in MS [720] have suggested their combination, i.e., an insulin-centered “diabesity” as a nuclear or main defining pathology of the syndrome [721]. Nevertheless, the importance of how diet, mainly concerning fatty acids and glucose (the main energy substrates), is handled, transported, or used in inter-organ relationships, and the direct implication or affectation of the connective tissue, help partially shift the focus of MS studies largely towards a type of connective tissue (CT), WAT, or white adipose tissue. CT is basically the live biological glue that holds the body together; it is deeply involved in the processes of growth, regeneration, energy handling, senescence, and defense against external aggressions. CT is pivotal for hold/support, movement of substrates and signals, and defense, i.e., the maintenance of overall homeostasis. However, this obvious transcendence is seldom analyzed such as within the context of MS, despite the direct relationship between WAT and obesity, a tangible proof of disorderly energy handling. Most of the questions affecting WAT in the context of MS (energy partition and storage, immunity, energy balance, etc.) can be easily extended to the whole CT, and its functions are the target of the alterations characterizing MS. Actually, adipose tissues are (sophisticated and often massive) forms of CT which serve, primarily, as energy security-oriented energy stores of lipid, or simply as ‘provisional’ (tending to become permanent) energy dumps because a large number of CT cells have evolved into adipocytes [722]. But, adipose tissue continues retaining most of the basic characteristics of CT, in aspects such as cell types, transport, signaling, repair, structure, and, especially, defense [723,724].

The CT dispersion and its variability in cells and extracellular matrices define a multipurpose, adaptive, and deceptively simple tissue, of which the main function is to hold, supply, transport, interconnect, and sustain the structures and components of the body [725,726]. In order to explain how MS induces a multipronged alteration in CT function, we need, first, to show a very general and simplified view of its different forms as found in human adults. CT is formed essentially by four different components, two of which are cellular in nature: A main cell type, adapted to the main function of the specialized form of CT (such as fibroblasts, adipocytes, mast cells, macrophages, and leukocytes).A number of other cell types, formed elsewhere and later migrated into the CT, but which definitive functions are largely carried out as part of the CT (such as different types of leukocytes: neutrophils, basophils, eosinophils, monocytes or lymphocytes) [727].

There are also two groups of non-cellular components: 3.A matrix of basal space-filling *ground substance*, a gel, pressure-resistant hydrophilic proteoglycan [728], which is a fundamental part of CT [729,730];4.Fibers, predominantly protein [725], from which collagen types are the most common and extended component [731,732]. Non-collagenous elastic fibers (mainly elastin) give structural flexibility and provide the tissue with a reticular holding-structure [733].

The stolidity of the ground substance and collagen are countered by elastic fibers, provide the mechanical balance needed for each site and purpose [726]. Table 2 shows the comparative properties of the different types of CT.

CT acts as the main homeostatic agent. It drives and holds the main blood vessels and surrounds, protects, nurtures, and maintains the connections of all the other organs and systems of the body [734,735]. It is a principal site for the recovery of lymph and other internal fluids released from capillaries and other structures, as well as a main recipient of the nutrients carried from the gut via lymph (and interstitial space) for (usually slow) distribution [370]. The functions of CT include the distribution (diffusion) of oxygen and nutrients, the maintenance of acid–basic and redox equilibria, as well as the maintenance of ionic content and functionality of the fluids bathing the cells and tissues [736]. The physical space of CT allows for the recruitment of phagocytic and other immune system cells, in the event of an insult, as part of the inflammation response. CT is a main site for the immune system’s defense elements to intervene [412,418].
ijms-25-02251-t002_Table 2Table 2**Main types of connective tissue of adult humans**.Type/SubtypeMain Junction(s)Main Cell TypeOther Cells *Matrix (Largely Amorphous)Fiber TypesLocation**loose**/areolarcover, bind, hold organsfibroblastsmacrophage, stem cells, adipocytes, leukocytesproteo-glycans and glycosamino-glycanscollagen loose, elastinskin, binding/holdingmost organs**loose**/reticularholding cells, filtering cells and lymphreticular cells (and phagocytes)macrophage, immune cells, leukocytesproteo-glycans and glycosamino-glycanscollagen (III)lymph nodes, thymus, liver, spleen**dense**/white fibrous[NBV]hard–delicate organ protection, heavy-duty mechanical workfibroblastsmacrophages, adipocytes, mesenchymal cellsproteo-glycans and glycosamino-glycanscollagen (I) packed, (elastin)fascia, duramater, spinal sheath, pericardium, cornea, tendons,periostium**dense**/yellow fibrousHard–fine repetitive mechanical workfibroblastsmacrophages, adipocytes, mesenchymal cellsproteo-glycans and glycosamino-glycans, chondroitinselastin, collagenligaments, vessel walls, vocal cords, lung, respiratory tract**cartilage**/hyaline[NBV]structural, shock-absorbing cartilagechondrocyteschondroblastsproteo-glycans and glycosamino-glycans, collagencollagen (II), (elastin)costal cartilage, nose, trachea, larynx**cartilage**/elasticvery flexible structural cartilagechondrocyteschondroblastsproteo-glycans and glycosamino-glycans, collagenelastinepiglotis, larynx, ear pavilion**fibrocartilage**[NBV]friction-resistant joint surfaceschondrocyteschondroblastsproteo-glycans and glycosamino-glycanscollagen, (elastin)meniscus, intervertebral discs**bone** (and teeth)maintenance of bone structures, support, armor protectionosteocytes, osteoblasts, osteoclastsstem cells, macrophageshydroxyapatite, dentin in teeth, proteo-glycans and glycosaminoglycans, collagenscollagen, elastin (mineralized)bones, teeth**bone marrow** hematopoietichematopoiesis, regenerationreticular cells, hematopoietic, and blood seriesstem cells, macrophagesinterstitial lymphcollagen, elastinbone marrow**adipose** whitefat storage, insulation, endocrine, immune control, regenerationadipocytesmacrophage, stem cells, leukocytes, immune cellsinterstitial lymphcollagen, elastinadipose tissue masses, organ interstitial fat**adipose** brownthermogenesis, heat diffusionbrown/beige adipocytesmacrophage, stem cells, leukocytes, immune cellsproteoglycans, glycosaminoglycans, glycoproteinscollagen, elastinbrown adipose tissue sites; often mixed with white**blood** (and lymph)oxygen, nutrient, and cell transport and delivery, coagulationerythrocytes, leukocytes, plateletsother immune system cellsabsence of solid matrix, just blood plasma (and its proteins)no fiberswithin blood vessels, heartNBV = no blood vessels across the matrix space; all interchanges with the blood are carried out from the periphery and or via lymph/interstitial fluid. * Plus migratory cells (largely carried by the blood, such as leukocytes) diffused into all CT spaces.


### 5.2. The Adipose Organ, an Energy-Handling Specialization of the CT

Obesity is universally considered to be a metabolic problem caused by the unnecessary and dangerous excessive accumulation of fat deposits (mostly TAG) in specialized cells, adipocytes [737], as well as in a large number of micro-droplets present mainly in the cell cytoplasm of several tissues and organs, such as the muscle [738,739] or liver [740,741]. Cell lipid droplets play important functions in the rapid availability of TAG for energy, defense [742,743], thermogenesis [744], and as short-term (or processing) energy buffers [745,746]. In adipocytes, lipid droplets play an important role in the transport of TAG and in their coalescence into the large storage vacuoles [747,748]. Adipose tissue is just a partially differentiated form of CT that primarily assumes function as the main body site for fat (energy reserve) storage. White adipocytes are differentiated, mainly from stromal cells and fibroblasts, into large cells mostly devoted to containing and maintaining enormous stores of TAG. Most of the mass of WAT is formed by mature adipocytes, but other cell types are found in globally higher numbers in the tissue. In line with what is expected from CT, these cells maintain a high metabolic activity and complement the metabolic and functional roles of the tissue [749]. Adipocytes (white) contain a large vacuole of lipid (essentially containing TAG) surrounded by perilipin and other frontier-function proteins [750,751,752], which help contain, maintain, and integrate the TAG depots in cell metabolism. Adipose tissue is largely formed by the accumulation of adipocytes forming the ‘white adipose tissue’, or WAT. The masses of WAT tend to differentiate and occupy body spaces in which their presence causes only limited functional problems (or where their mechanical/thermic buffering becomes an asset), and are correlated with the massive presence of intracellular fat depots and the presence, but also of interspersed cells or small groups of WAT cells within a number of organs and tissues, such as the muscle: intramuscular [753,754,755], pericardial [756], perivascular [757,758]; other specialized tissues, bone marrow [759,760], or mammary tissue [761]; as well as surrounding them for physical/thermal protection, nurture, and regulation: perigonadal [762,763], subcutaneous [764,765,766], perirenal [767], and mesenteric [768,769] adipose tissues. The location and close relationship with specific organs, sharing contact and direct metabolic and humoral-regulatory connections, result in a high level of sub-specialization in both cell composition, depot size, and metabolic activity [770,771,772]. There has been considerable discussion on the pathogenic significance of their gross excess of energy, which is finally converted into adipocyte depots in the form of defined WAT masses, and of groups of adipocytes interspersed with other tissue cells in organs and tissues. In addition, there is the direct accumulation of lipid droplets (located in the cytosol of tissues inclined to use them for ready-use energy. The accumulation of fat in adipose and other tissues, however, follows a fairly (body) uniform pattern, irrespective of location and specific depot size [773].

The most-studied specialized types of adipose tissue are, perhaps, those related to thermogenesis [774], both because of their heat-producing function, maintaining body heat and helping maintain temperature stability under exposure to cold, as well as the wasting of excess energy substrates as heat through their wasteful (non-productive) oxidation. Thermogenesis is an important factor for survival against the cold [775,776] and a critical factor in the maintenance of the body’s energy balance [777,778,779]. The importance of this outlet of energy towards the production of heat is more important for small homoeothermic animals dependent on the maintenance of body temperature, high metabolic rates, and the added need to limit the size of cumbersome WAT depots [775,780]. For humans, these processes are also important, but in this case, the main problem in most environmental conditions is the maintenance of an adjusted level of reserves and not so much the loss of heat [781,782]. The relatively higher body mass of humans (and WAT insulation capacity, compared with those of rodents) is critical to achieving these objectives [783,784]. In any case, the thermogenic mechanisms are essentially adaptive [782,785,786], but their regulation is multifactorial and not completely understood. It is assumed that thermogenesis is carried out by a number of tissues with high metabolic activity, i.e., by incorporating, in part, the residual metabolic energy loss of futile cycling of exergonic combined reactions [787,788], or simply shivering [789,790], ineffective processes in liver [791,792] and skeletal muscle [793,794], and, especially, the uncoupling of oxidative phosphorylation in the mitochondria of, mainly, brown adipocytes [795,796]. These are a specific type of cell found in WAT masses and/or in isolated depots around core organs [775,776,780] or vessels needing special protection against the cold and hypothermia [797]. The beige adipocytes also participate in adipose tissue thermogenesis [798,799] and share, with brown adipocytes, the abundance, effectiveness, and regulation of mitochondria [800,801,802] (and the brownish tell-tale color of their cytochromes) with other highly oxidative tissues (such as hepatic parenchyma). In brown adipose tissue (BAT), the presence of a membrane G-protein, UCP-1 [803,804], allows for the unhindered return of protons from the cytosol to the mitochondrial matrix, thus bypassing the directional ATP synthase systems and, consequently, ‘losing’ their gradient-derivable energy as heat. The browning of WAT is an important process—clearly observed in rodents [805,806,807] but also present in larger animals such as humans [808,809]—that allows for the development of uncoupling cells within WAT to contribute to the conversion of excess 2C (and 3C) energy into heat, as in BAT. In both adipose tissue types, this process goes against the ‘rules’ of energy preservation and has difficult application because of the numerous metabolic protective circuits and restrictions, thus requiring a transformation/change in the whole-cell organization to become functional.

Nowadays, adipose tissue has become a widely analyzed subject of MS-related disorders, mainly diabetes and obesity, often because of its non-physiological excess rather than for adipopenia, which is usually linked to nutrient deprivation and the pathologies of scarcity and cachexia. This interest is slowly changing our perception of WAT from an inert energy dumping site to a tissue (organ) sporting many more active functions in addition to the handling of TAG depots, such as the conversion of 6C to 3C, an active metabolism of 3C units (i.e., producing lactate [810,811,812], glycerol [813,814], or alanine [815]), and a less-known ability to metabolize amino acids [816,817,818]. This high activity in relation to a relatively limited cytosol mass is compounded in WAT by a normally low pO_2_ [819] and relative mitochondrial scarcity, both in agreement with a marked glycolytic shift of its 6C ↔ 3C metabolism, which allows for its operation with a markedly low consumption of oxygen [820]. Cultured adipocytes maintain their largely anaerobic metabolism of glucose even under perfectly aerobic conditions [821], a peculiarity shared by the non-adipocyte cells of WAT, which globally exhibit a relatively higher lactate production from 6C than the adipocyte fraction [749]. The in vivo varying pO_2_ values, however, have a considerable importance in the insulin regulation of glucose uptake and glycolysis, at least in rodents [820].

The overall mass of WAT (and, to a much lower extent, BAT) in an adult human is considerable (in the range of 8–20% BW and upwards, to higher incapacitating percentages); its rather uniform behavior with respect to substrate handling as well as energy storage and disposal have spurned the definition of the combined body mass of WAT as a disperse but well-defined adipose organ [822,823]. This clever conceptual jump has helped us to better understand the role and functions of WAT. As an organic unit, the explanation and understanding of many of its regulatory mechanisms can be reinterpreted, i.e., the internal shifts in metabolic orientation derived from obesity [824], browning [825,826], and the transfer of mitochondria or high-mitochondria-containing cells that help modulate the responses of WAT to changing metabolic conditions [497,827]. WAT performs other organ services in addition to its main implication in body energy metabolism [828,829,830], such as providing stem cells for repair and reorganization [831,832]. It is also a site for steroid hormone interconversions [833], largely affecting DHEA [834], corticosteroids [835,836,837], the metabolism and aromatization of androgens [838,839,840], and estrogen interconversions [841,842,843]. Adipose tissue is a prime site for the anabolic action of insulin [844,845,846], BAT for catecholamine-related energy-mobilizing (and wasting) effects [847,848]. The active production of cytokines (adipokines) [849,850] and its intervention in the regulation of the organs/structures that the tissue covers, packs, and protects [851,852] support the concept of WAT as a specialized disperse ‘organ’. Its important integration of defense systems [853,854] and the uncanny “association” of WAT-originated cells in the development and sustenance of many types of cancer [261,262,855,856] prove that we have a long trek ahead to fully understanding the many facets of the adipose organ.

In any case, all these peculiarities that we recognize and analyze in the adipose organ are just part of CT’s general functions of organization, connection, provision, protection, support, service, and, fundamentally, homeostasis-preservation. The old (albeit partly surviving) concept of WAT being a rather ‘inert’ tissue can no longer be maintained, since its metabolic activity is high when compared with most body tissues in relation to their ‘live cytoplasm’ vs. total tissue mass [749], i.e., when discounting the mostly inert (anabiotic?) mass of stored lipid. The endocrine/paracrine activity of adipose tissues (i.e., WAT, but also BAT), and the metabolite, regulative systems, and coordinated function relationships between WAT and the liver/main gut microbiota, the brain and other organs point to a much more deep functional interaction between them, as well as with the energy, nutrient availability, and nutrient fate, either through direct and/or bidirectional crosstalk [857,858,859,860] or threesome interventions with the gut [861,862], bone [863], or other tissues. There is a functional continuum between CT and WAT (logical, since the latter is a ‘specialized’ part of the first), with only adaptive modulation to maintain function where needed even in the context of severe energy dysbiosis. Again, the comparisons with cancer come to mind, since the critical help provided by the WAT (or perhaps the CT roots) cells, including the heavy load of energy they carry, plus the plasticity of its stem cells and fibroblasts in its neoplastic success, helps explain the ‘servitude’ or secondary collaboration of CT (or only WAT?) with respect to all other tissues it helps to protect and maintain in due order [261,855]. This ‘service role’ is maintained even under conditions of deeply altered functionality, as in MS. In fact, the scarce data available suggest that the relationship between MS obesity and cancer is more a consequence of obesity as such than stemming from the global MS mode, i.e., [MS → ↑ obesity → ↑ neoplastic growth], implemented largely via cell recruitment [858,864].

These considerations hold open a number of possible avenues for the treatment of MS based on the weight of the obesity dysfunction (both at the start of altered metabolic manifestations and as a consequence of the massive size of “impossible-to-use” reserve depots).

## 6. The MS Disease

### 6.1. Pattern of Development of the MS

The MS, normally, has a slow, progressive appearance in parallel with aging, but it is generally agreed that it is a pathologic state typical of maturity, i.e., it is usually detected in mature adulthood, but expands and intensifies its manifestations well into advanced age, a timespan which is often cut short because it strongly increases mortality [58,865,866]. The main factors favoring MS development and maintenance are as follows:(a)*Age; senescence*.Typically, the first markers of MS appear early in men. Lore informally establishes the fourth decade of life as when (essentially men) become aware of ‘becoming old’ through a number of functional, psychological, social, and health shortcomings, often eliciting drastic worsening changes in their psychology and activity [867,868]. This phenomenon also occurs in women, albeit being more diluted over their mature years because of higher social environmental pressure and also higher resilience [42,47]. In women, the ‘discovery of aging’ is often acknowledged later, close to menopause. In both genders, after these crises comes an interlude for acceptation and adaptation to the altered concept of their life/personality (or for its negative psychological blockage). Advancement of the MS condition, especially that of obesity (which can be easily detected/commented on by others) [869,870] and liver steatosis [871], are also accompanied by stress [872], and then by other MS components, that elicit additional consequences of (or caused by) changes in social behavior [43,873,874]. These changes may bring in a new phase, the conscience of bearing a progressively severe crippling (and eventually terminal) disease, can be socially devastating for many patients. Often, this awakening does not take place this way, because one of the main MS spectrum diseases is the (practically unique) focus on health-menace attention; it is ‘the disease’, and thus the other manifestations are taken as subordinate, not as syndromic partners of generalized inflammation consequences [875]. This point of view is in part also shared by a high number of health professionals, and may result in the establishment of conflicting therapeutic priorities that should be carefully pondered when treating a polymorphic pathologic state, largely relying on damaging, ineffectual, and expensive polypharmacy [876]. At this phase, most disorders are already treated in a ‘sustaining’ or ’palliative’ way because of a lack of effective alternatives, i.e., treating the conditions as best as possible to gain time and improve the living conditions of the patients. Unfortunately, this humane and ethical way forfeits real healing due to the intertwined nature of the illnesses implicated [28], and their extension and transcendence (e.g., sleep apnea [325,877], type 2 diabetes [878,879,880], and most MS-related heart and circulatory conditions [275,288]). There are, too, complicated avenues for treatment that are not always well defined, applicable, or even available to most patients, such as depression [881,882] (not even agreeing on its relationships with MS [883,884]), osteoarthritis [238,239], psoriasis [126], or even severe obesity [885,886], which is the leader of medical, social, and economic studies (and failures). This situation continues until tumors, dementias, aggravation of a principal disorder, or, commonly, vascular or heart failure cuts a life, between late maturity and old age, after a course/curse of (often) many years.(b)*Sex; planned obsolescence*.Women constitute the bulk of patients with diagnosed MS [887,888], but men often present more severe (and lethal) manifestations of MS [866,889]. The superposition of menopause, senescence and MS in women, compared with the earlier onset of severity in men [48], somehow obscures the choral effects of MS [866,890]. The higher capability of women to limit the ravages of disordered lifestyle involves a careful, often restrictive, choice of food [891], together with higher psychological fortitude [892], an ‘intimate’ relationship with E2 [893], higher X chromosome reliability [894,895], metabolic resilience, and lower MS-related cardiovascular consequences [48], and mortality rates than men [866,896]. In men, MS and most of its morbidities are related to low blood T [897,898,899]; a marked lack of enough T also results in lower dihydrotestosterone effectiveness [900] and a parallel decrease in E2, which [95,124] results in a further lack of control of the insulin–glucose system [898,901] and a decrease in the flow of 2C through the Krebs cycle (Section 4.2 and Section 4.3). This consequence is a marked incapacity for processing the excess 2C induced by diets rich in 6C, but even affecting diets that are not hyperenergetic at all [902]. The deficit in T is also crippling because hormone production is not sufficient to sustain an adequate mass and turnover of body protein [903,904,905], which may eventually result in overall protein loss and sarcopenia [50,295,906], a characteristic of advanced senescence [907,908]. A deficit in T (controlled by the brain, via the hypothalamus–hypophysis–gonadal axis) at mature age results in a marked decrease in sexual interest, arousal, frequency, and effectiveness of sex [909,910,911], but also a progressive loss of muscle and whole-body protein [912,913] (including that of the heart [914]), higher insulin resistance [915] plus lower capability to oxidize 2C (thus favoring fat deposition) [916,917,918], depression [919,920,921], and deep hormonal disturbances [50,922,923]. The consequence is, usually, an earlier than expectable demise. These extended effects, at least in men, may be compared to those of a programmed obsolescence [201], which shortens the life of individuals of the species when they are no longer “needed” by the group, thus saving resources for the rest. In fact, senescence is exactly that, a genetically planned mechanism to cull the individuals which no longer reproduce, take care/protect the pregnant, younger/elder, or obtain food, defend the tribe, etc., but consume otherwise (collectively) needed resources. Women’s lives endure a stop to ovulation (and related hormonal milieu) when their remaining lifespan (and physique) cannot safely guarantee their contribution to sire new individuals (neither, the continuous provision of care and protection to them). However, (older) women’s resilience is (apparently) relied upon by our innate biology rules, by allowing them to continue help caring... for other women’s children; they already have almost a lifetime of training, retain the capacity to carry on these tasks, and commonly live up to fairly more advanced ages that men [924,925,926]. Anyway, some sort of gender-related planned obsolescence is probably set in our genes, and may be detrimental for men, at least in the question of survival/longevity (via resilience?). The question of whether MS is an instrumental agent for the implementation of a gender-related planned obsolescence, remains, thus, a plausibly probable (but so far unprovable) hypothesis.(c)*Diet and food security*.The direct relationship of MS with the problems caused by excess dietary energy supply, as well as the effects of inadequate structuration of nutrient availability, have been described and explained in Section 4. The relationship between the development of MS (liver steatosis, type 2 diabetes, and obesity, at least) may be traced to the repeated use of diets which cannot be properly handled under the metabolic (and, especially, regulatory) conditions affecting individuals with (or prone to) MS [927,928,929]. The nexus of the insulin–glycaemia system with the estrogenic regulation of 2C disposal (and, by defect, of enhanced synthesis of fatty acids) supports a clear endocrine basis for MS [133,152,192], more so when we include the compounding deleterious effect of insufficient availability of T cited in point b). Glucocorticoids tend to antagonize testosterone in the preservation of protein [99,110] and facilitate hyperglycemia against the reverse effect of T [930,931,932]. The equilibrium of these classes of steroid hormones is essentially controlled by the brain, thus adding another level of regulation (nervous) to the core mechanisms of energy partition. The brain also has an important role in the control of appetite [160,161], including complex mechanisms based on energy function markers, but also on hedonic signals and memory [933,934,935]. Another factor related with food use is the control of stress caused by the imprecise availability of food, food security [8,936,937], which has not been yet made universal for humans, but which, from the ‘human’ perspective of MS (described in Section 1.1), the anxiety for obtaining the next meal is a condition relegated to the rest of animals, since human ingenuity has worked over thousands of years to close on this ultimate level of security. Unfortunately, the fight and strife between human groups has prevented, so far, a complete universalization of this primary objective. Food insecurity is a main cause of stress [938,939], which represents a corticosteroid-dominated regulatory milieu [940,941] in the context of the modulation of feeding by the brain [942,943,944].(d)*Social enviroment and its effects on psychology and behavior*.Social constrictions affect the consumption of certain types of foods, a signal of identity, tradition, or secular availability (including religious and social habits and mandates) [945,946,947]. These are aspects deeply affected by the availability of food [948,949], its variety [950], and the regular supply affecting the food security aspects presented above [951]. The epidemiology of MS shows that, in a few generations, a society may often pass from a survival stage, under the permanent danger of famine, (nutrition transition) to high food availability [952] and gluttony-overeating [953], helping to induce a massive development of obesity [954,955]. Probably, the sociobiological roots of these (social evolution) sudden shifts may lie in ‘epigenetic hunger’: memories and functional/behavioral patterns acquired from prenatal experiences or transmitted from previous generations [956,957]. The relative coincidence of increases in MS expansion in countries that have recently endured harsh stress and starvation conditions hints at learned overeating as a precaution [953], but also at overfeeding of children, as is the common situation of proudly showing-off chubby babies (linked to a positive view of children obesity) [958]. These attitudes have been in use as ancient recipes for survival, and remain clearly marked in our genetic background. Nowadays, in most communities and close-knit groups, any meeting justifies excess eating [959,960]. Food sharing promotes feasting, overeating [961], access to more varied foods [962], and is an element of social unity, hospitality, friendship, and positive relationships. As such, this universal—and quite diverse—behavior is a critical part of social groups’ cohesion via tradition and heritage. In many social events, the excesses (aka “justified” exceptions) in energy intake (and loose behavior) are part of the expected (i.e., “normal”) behavior. Nevertheless, socialization should not become a disrupting factor in regular eating patterns, especially when the risk of famine is low and that of MS is skyrocketing.

Figure 8 shows that the factors described above fit within the lines of metabolic regulation of nutrient partition and the cumulus of diseases constituting the MS. It is generally accepted that genetic background may favor the development of MS (or that of its main components: obesity, type 2 diabetes, hepatic steatosis, atherosclerosis, AHT, etc.). It is, again, generally accepted that the brain (via psychological circuits and/or neuronal/neuroendocrine actions) controls the whole set of hormones and metabolic paths affected by MS, including widely extended diseases of yet uncertain understanding, such as depression. Aging is a process that MS mimics, in part, by sharing paths and control systems. Diet and its extreme variations in components and timing have already been exposed above. Then, the direct regulatory systems affecting nutrient partition: the insulin–glycaemia system and its main controllers: T-E2 (and GC as countering element), ultimately determine the development of MS. The graph is a simplified depiction of a model which is much more complex, but the basic conditioning blocks are the same.

### 6.2. MS as a Branching Monophyletic Disease

Figure 9 shows a schematic view of the main relationships that have been established between the different disorders and diseases described as part of the MS. The coincidence in paths, relationships, and grouping of pathological traits should not exist if the different related pathologies were simply ‘running together’. But, the objective of Figure 9 is different, since it reflects the directly paired relationships observed between the different symptom/disease/disorder parameters shown. For most of these metabolic and regulatory relationships, there is a considerable amount of proving information already available, especially in the references to insulin function and the maintenance of glycaemia, together with the closely related obesity.

The meaning of inflammation as an altered physiological (and biochemical) state is at the center of MS, largely related to CT [18,28], which energy depot function—as adipose tissue—is widely altered during the development of the syndrome, since obesity (in fact, ectopic non-physiological excess fat deposition) is one of the main signs of MS (Table 1). The mass effect of the extra body (fat) weight results in alterations of respiratory [963,964], heart, and vascular function [965,966]. Excess energy, largely from carbohydrates and lipids, favors the development of liver steatosis [967,968], as well as the release of an altered pattern of blood lipids, exporting the problem to all other organs and systems [969]. The corresponding MS components are intertwined, causing concatenated failures which are recruited or incorporated to the pathologic wave because of their direct interdependence: biochemical, functional, and regulative. These MS alterations can be largely traced to problems of energy partition (Section 4.2 and Section 4.3) caused by excess (and untimely) nutrient availability. But, liver dysfunction is also the concomitant consequence of an altered insulin function (insulin resistance) [91,970], also related to diet [971]. Consequently, the glucose homeostatic systems cannot maintain well-regulated glycaemia and insulin function [879,970]. They are in themselves dependent of medium-term control by E2. which effects have been observed directly [152,972]. The availability of E2 is also directly related to that of T, its levels also affecting hepatic steatosis development [973,974].
Figure 9**Causal interrelationships between the different pathologies that integrate MS.** The core of the MS is the establishment of an inflamed state in the CT–obesity relation (marked in the graph with a wider connection line). The different disorders, symptoms, mechanisms, and diseases are marked by rectangles. Titles in blue represent the organs or tissues, and the pathologies are in black. The “mass effect” stands for the consequences of obesity directly attributable to an increased body or organ/tissue mass (i.e., circulatory resistance, higher cardiovascular capacity, oxygen demand, and energy cost of movement). Up arrows (↑) represent increases, and down arrows (↓) decreases. Abbreviations: T = testosterone; E2 = 17β–estradiol, GC = glucocorticoids; AHT = arterial hypertension. Green arrows represent known causal or mechanistic interrelationships between the pathologic traits within the context of MS. The brown arrow represents a possible causal relationship, so far not proven.
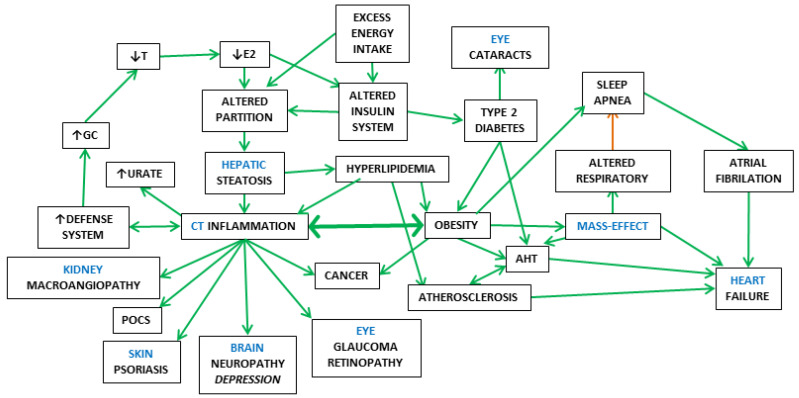



Diabetes (and its consequences of cataracts [975], macro- [976] and microangiopathy [977], AHT [280], etc.) is directly related, in the context of MS, to alterations in glucocorticoid function [978,979], partly counteracted by T [189] and E2 [190,980]. Diabetes is correlated to excess energy availability [981], altered partition [982], hyperlipidemia [983], and obesity [984]. Tissue inflammation is also related to the ubiquitous presence of CT [725] and to its function as a main communication highway, via blood, lymph, and the maintenance of intercellular matrix and its homeostasis [230,985]. Inflammation also activates the defense system cells [986,987] and the regeneration/growth that is established mainly via stem cells [988], fibroblast modulation [989], and phagocytic debris- disposing-of/digesting cells [372]. The implication of MS (or obesity) in a number of types of cancer is well known [320,990,991] and related to the tumor-hijacking of cells, with the direct use/collaboration of WAT/adipocytes [855,992], stem cells [264,265], fibroblasts [993,994], and even macrophages [266,995] to fuel, protect, and help metastasize tumors.

The extension of altered function in CT affects most tissues and systems, such as: (a) the cardiovascular [996]; (b) the eyes (retinopathies [307], glaucoma [337]); (c) the skin (wound healing [399], hypersensitivity, psoriasis [126]); (d) the kidneys [305]; (e) the endocrine system (a cause (or consequence) of hypotestosteronemia and of its effects) [201,899], but also including the polycystic ovary syndromes [997] as well as the main hormone-related cancers (endometrial, breast, prostate) [998,999]; (f) the muscle and bone are also deeply affected, with muscle withering towards sarcopenia [87,1000] and, secondarily, frailty [1001]. Bone–cartilage structures also show marked dysfunctions/damages, affecting the extracellular matrix and overall function, such as osteoarthritis [736]. The brain is even more markedly affected, first because of altered steroid hormone equilibrium and overall endocrine control [163,166]. There are implications on the brain’s structure [113], affecting cognition [1002,1003], memory [1004], and resulting in a number of psychological and neural dysfunctions [113], including depression [881,882] and changes in self-image. T levels [921], glucocorticoid stress [940,942], and lifestyle [55] are also altered by MS. Senescence, neuropathies [114], neurodegeneration [1005,1006], and dementias [270] are also interrelated with the cumulus of disorders presented in Figure 9. In all of these cases, the mass of evidence accumulated on the interrelation of the clinical parameters related to the different MS components is already considerable (albeit incomplete), establishing solid direct relationships of probable causality/consequence between them.

These converging relationships hint to common pathologic origins (namely: aging, CT inflammation, and, at least, T availability). Their combined effects should also be interpreted together with the known mechanisms of the regulatory control of genes through the nervous system, i.e., organization structures [1007], rhythms [1008,1009] (plus endocrine regulation), constituting a block of *internal* factors. On the other hand, as the main *external* factors, we should include diet, environmental factors, stress, and lifestyle/social environment.

Nonetheless, the most striking non-compliance of the long list of acknowledged MS components can be found in one of the older members of the MS club: obstructive sleep apnea or OSA. This is probably the most common form of sleep-related apnea, in itself a heterogeneous disorder that responds to different origins (idiopathic, neural, anatomic, etc.). The close associations of OSA with obesity [324] and insulin resistance [327] have eased its incorporation into the MS cluster, but its patent heterogeneity makes it difficult to attribute a single cause to this widely extended disorder. Its epidemiology [1010] and consequences are, however, better known, with a close relationship with the irregular discharge of catecholamines as response to the extended/repeated periods of hypoxia [877]. This situation alters the heart’s rhythm [333] and elicits cardiovascular consequences [332], in addition to the problems caused by disordered sleep [1011] and respiratory complications [1012]. These problems have a multiplying effect when translated to their effect on behavior [1013]. There are no reliable hints as to how these relations and consequences are derived from other common, known pathologies, of which roots may also be heterogeneous. There is a general, albeit unproven, assumption that OSA may be, in part at least, a consequence of the unresolved inadequate anatomical design of the upper respiratory tract for maintained efficiency when the erect upright human posture for daily activity is changed to horizontality for rest and night sleep [1014]. This is probably compounded by muscle relaxation and so far unknown effects of body fat distribution in the neck [1015]. OSA is currently treated with hygiene and palliative means [1016]. It is commonly maintained within MS (it is, nevertheless, syndromic) as an associated disease, but not as a fully integrated component of MS when it is considered as a monophyletic disease.

Despite considering the important exception of sleep apnea, the development of most MS pathologies can be traced along known and shared common origins, direct relationships, or shared mechanisms. Evidently, the adscription of inflammation to the core of physiological and biochemical mechanisms has considerably advanced both our depth of knowledge of MS and its biological significance [18,1017]; however, the traditional organization of diseases and disorders by way of the physiological system affected weighs too much to be readily integrated in wider visions of complex pathologies, such as MS.

### 6.3. Conclusions and Perspectives

The overwhelming pathologic cause–effect relationships between the components of MS prove that they share an alignment of causal factors and a common intertwined origin, but the different strengths of its components enhance its variability and the eventual assignment to a main cause (a ‘primus inter pares’ disease). The unicity of MS and the interrelationship of its components have been repeatedly analyzed and proven to exist. However, there is not yet a consensus on the identity of the “primary” trigger mechanism. We can safely assume that even if we do not accept the ‘planned obsolescence’ hypothesis, we can forfeit the redundant ‘planned’ (since all gene-related mechanisms are—conceptually—genetically planned, in line with what we know of MS genetics [33,35,1018]) but the obsolescence case remains, in part, unexplained/unjustified by non-steroidal hormone alternative mechanisms. In addition, there are partially superimposable mechanisms to compare (i.e., menopause) which produce deep alterations: functional, metabolic, and behavioral, but which do not share the decimating effect observed in mature and old men. The most plausible “primary” causes or starting points to analyze the situation may be the following:(a)*Type 2 diabetes and obesity*.So far, this is the (apparently) most medically internalized mechanism used to explain the development of MS. In its favor, we can include the pivotal role of insulin on carbohydrate metabolism, energy partition and supply, the use of glucose, and the 6C–3C relationships to sustain a large part of the body’s energy needs. It is directly related to the insulin-favored deposition of excess energy as TAG, and the correlation of obesity and insulin resistance (and inflammation) is strong. In most cases of MS, insulin resistance is present, often accompanied by type 2 diabetes and obesity [135,870]; in fact, most people with MS are directly treated for diabetes and obesity. Type 2 diabetes and obesity are mainstays of MS, and are related to most of its other components and consequences, including their incidence in cardiovascular disorders. Nevertheless, type 2 diabetes also exists as a more-or-less standalone disease [1019,1020], and obesity may be (albeit in a small proportion) a consequence of other disorders [1021]. There are common cases of obesity (metabolic) without excess fat deposition (anatomic), as is the case of “normal-weight obesity” [1022], an oxymoron indeed, of which existence is not yet fully acknowledged [1023,1024] despite the finding of a number of complications affecting the ‘normal-weight obese’ [1024,1025,1026]. The timing may be important, since clinically defined MS is often declared later than obesity and/or diabetes. The main factor against a simple sequential hypothesis is the diverse (and complex) pathological signature of MS compared with the more lineal development of diabetes and obesity-related complications. Their causes (genetic, endocrine, dietary, etc.) are also often traceable. In any case, the question of them inducing a trigger effect remains, since the type 2 diabetes/obesity tandem appears, more often than not, as a harbinger of full MS [1027].(b)*Hypotestosteronemia*.In general, MS can be traced (or found to be parallel) to low T availability in mature (or old) age [94,899]. T has been intensively (and almost exclusively) related with sex (as also occurs with E2) [1028], but, nowadays, its metabolic maintenance function is fully recognized [201]. The need for E2 to facilitate the initial phases of energy partition is known but not yet fully integrated in clinical treatments [133,195,1029], in part because of prevalent bias against the use of ‘sex hormones’ in the treatment of metabolic (not explicitly sex-function-related) disorders [152]. The development of this line is just one step ahead of the preceding case in relation to medium- and long-term regulation: low T results in low E2, which then cannot maintain a tight control of the insulin–glycaemia system regulating dietary energy partition. The main real argument against this line of thought lies precisely in the wide extension (and roots) of ‘sex hormones’ as a ‘group’ of hormones identified solely with the gender-related regulation of sexual/reproductive functions. Consequently, both androgens’ and estrogens’ functions are contemplated mostly as gender-differentiating agents, minimizing the specifically metabolic regulatory roles shared by females and males outside of the additional limitations of age, sex, and maintenance of energy metabolism. In sum, sex is sufficiently transcendental (the future of the species depends on it) by itself to also include the control of energy metabolism. For a large portion of clinicians and basic scientists, energy metabolism is already controlled by appetite, insulin, and a large (and growing) number of additional known and largely independent mechanisms. However, the need for E2 for male reproduction and T for that of females, androgen and GH for body protein maintenance, and estrogens (and androgens) interacting between them and with glucocorticoids (and many other regulatory molecules), constitutes a whole intertwined set of medium-term hormonal control processes that affect both male and female individuals from cradle to grave. Additionally, both T and E2 are needed in both sexes for effective reproduction, with androgens eliciting a ‘male’ phenotype and ‘estrogens’ favoring a ‘female’ phenotype. Even in the absence of truly sexual development (or outside of the ‘reproductive window’, in practice including about half of a lifetime or less), androgens and estrogens keep maintaining their metabolic homeostatic functions (albeit impaired by eventual insufficient secretion).

The nature of extended low-key stress (including a predominance of glucocorticoids [1030]) may modulate the development of a number of MS entities, being a significant part of MS because of its direct relationships with the defense system and inflammation [1031,1032,1033]. Other behavioral and lifestyle factors markedly influence the development and maintenance of MS, including early food deprivation, as proven by the sequelae of famine affecting whole populations decades later [1034,1035,1036]. Evidently, the conditions of harshness, vs. those of excess and abundance, with the necessary adaptations for survival have an important collegiate effect on the development of MS [953,1037]. Chronic stress situations [68], such as those found in many job environments [1038], have been found to elicit the development or aggravation of MS. A similar situation can be found even under adequate feeding when diet composition and sufficiency shift largely to excess. In most cases, diet is a fundamental factor in the development of MS, but its possible incidence as triggering factor is difficult to justify because a perfectly normal human should be able to sustain (and contain) the sporadic aggression of excess energy in the diet (at least for a time) by simply applying the inbred neural-regulated mechanisms of food intake that humans share with most animals [1039,1040] (e.g., eating less and/or increasing energy expenditure). The type of food consumed may not be, by itself, a sufficiently powerful reason to elicit the appearance of MS; this does not exclude the development of some of the diseases associated with MS, but hardly that of the full MS spectrum.

There are considerable gaps in our knowledge of MS, but the pace of advance is increasing. However, the question of translating the meaning of ‘healthy lifestyle’ to *living with* MS is far from being solved in practice. The widely used notion of ‘healthy’ (products, foods, behavior, etc.) is massively diffused and praised, making it almost universally accepted and keenly followed... Unfortunately, the verbal declarations of ‘healthiness’ are seldom justified with hard facts, and are neither suitable for real-life settings. Worse, most of their beneficial effects remain unproven. In line with this lack of actualization, the guidelines and counsels given by numerous authorities in reference to the MS constellation of pathologies are largely limited to general terms and have hardly changed over the last decades, in spite of our real advances in knowledge during that time. Too much valuable information is lost along the journey from published paper to practical application, and thus does not help to actualize the large interludes of inadequate information, carried on from the past, that distort our explanations, therapy, and research alike. The importance of diet as a ‘main’ cause of MS, pathogenic agent, and guilty part of almost every one of it effects continues being the subject of an enormous mass of studies, conducted using conceptually repetitive schemes and also limited testing variations and experiences, dealing with foods, nutrients, conditions for eating, timing, taste, and often using insufficient, incomplete, and essentially crude set ups. The few recommendations passing through these barriers are often diffused out of context and commonly extended to conditions widely different from those used to obtain the information, commonly jumping conclusions from mouse to patient and then further generalizing... In any case, the worst problem to solve remains, probably, in how to deal with the enormous mass of information gathered so far, published in often inadequate conditions of research and diffusion, compounded by the spurious influence of too many interested parties (often able to veto) and the absence of any type of organization of the materials actually available, i.e., when not restricted in publication and/or diffusion by the skyrocketing costs of research, publication and access, and other elaborate types of constrictions (but all of them largely devoid of new ideas).

As to the possibilities of effective treatment, the present-day piecemeal approach to the most lethal (or of those treatable) pathologies should be widened, in order not to induce (or sustain) unwanted crossed iatrogenic complications or undesired regulative alterations. The main problem is that many of these medical actions may be, essentially, marginal with respect to the actual causes eliciting MS, and their sought effects are mostly palliative (if any). However, the application of time-adjusted substitutive hormonal treatments (at present, basically using T), the reconsideration of polypharmacy as a standard (often iatrogenic) procedure for most patients, and a much-more-adjusted diet (in line with the *known* biochemical pathways of substrate utilization, their compartmentation, and regulatory mechanisms) may help improve (or, at least, not worsen further) the efficiency of energy partition and substrate handling, thus depriving MS of a main aggravation factor: disordered dietary nutrient handling. The limitation of available excess of energy may dampen the multiplicative recruitment of related pathologies and, thus, decrease the severity of the disease and help limit its ravages. This is a sensible line of work, but it needs, essentially, a deeper knowledge and a wider scope of the disease to succeed in the treatment of MS. This last point requires an exercise of adaptation to disorders affecting multiple interconnected systems, a question which compounds the need to digest (and translate into practice) enormous loads of information (and pick the grain from the enormous mass of chaff). 

Probably the best way to cope with the already present (and growing) epidemics of MS will be that of enhancing the creation of evaluation and treatment groups including a diversity of health-practice specialists. Along this line there is, already, a growing number of health centers that have specific clinics or organized teams specialized in MS. They include trained medical specialists working in group related to all the specialties dealing with MS outshoots, using well-defined protocols and guidelines, and maintaining in close contact with external analysts and research hubs. The actualization of medical teams is a critical issue both for the complexity of their tasks, the quality of their specific work, and for the effectiveness of their treatments. This last point is probably one of the most difficult to tackle, since the requirements include, at least: (a) a change in paradigm, (b) sufficient funds, (c) initial formation, cohesion, and constant updates (including two-way contact with knowledgeable scientists) of the whole staff, (d) true team work; and (e) an enormous effort in organization and logistics, including an executive, truly ethic, and knowledgeable direction; access to sturdy and tested new analytical techniques; readiness to adapt/or change established routines; rapid contacts for consultations; and a dynamic web of information, formation, assessment, and potentially personalized care of patients. This maybe seem an utopic dream, but the problem is there (to stay and harm) and keeps growing. The disease is complex, and the extremely few people actually fighting it need to organize and work (against odds) with the best minds, knowledge, and materials (devoid of red tape and oversized egos) to stand, at least, an opportunity to succeed.

In any case, we have advanced enormously from the first hints of a serious complex condition linked to affluence and initially defined allegorically as the *X* (for unknown/mysterious) syndrome, and later as a *deadly quartet,* alluding to its, then, four main (deleterious) syndromic components. Nowadays, we are already facing an even deadlier challenge (i.e., *the quartet* grown to a *full symphonic orchestra with choirs*), of which expansion in morbidity (and enhanced mortality) continues, overcoming our intended (limited) efforts to stop it, inducing fear, uncertainty, and suffering to millions, *and* decimating our societies.

Nevertheless, at least, and despite the growing odds, we should retain hope, because we are beginning to understand the causes, means, and mechanisms of MS; a necessary step for its effective prevention, true control, and, eventually, generalized cure.

## Figures and Tables

**Figure 1 ijms-25-02251-f001:**
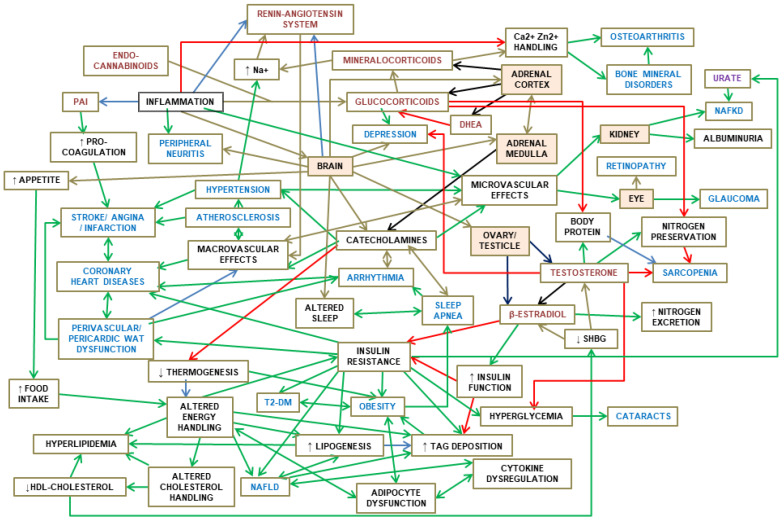
**Main relationships between the most common MS-constituent diseases/disorders.** The graph is focused on the core of energy metabolism and its regulation in humans and the direct relationships between the different MS traits/pathologies, with regard to the main known mechanisms of endocrine/metabolic regulation. A few organs/tissues are shown as small rectangles with pale tan background. Pathologies and groups of disorders are shown in blue; main regulatory agents are presented in purple; and metabolic changes, effects, or processes are shown in black lettering. Inhibitions or blockages are represented by red arrows, activation or enhancements in green, and more complex regulatory or control effects are depicted with brown arrows. Black arrows show the direct production, secretion, or synthesis relationships. For clarity of the graph, not all known interrelations have been represented. Small arrows in the text (↑ and ↓) represent the modulation up or down of the parameter. This scheme is only a simplified (incomplete) presentation of the relationships hinted at in the text.

**Figure 2 ijms-25-02251-f002:**
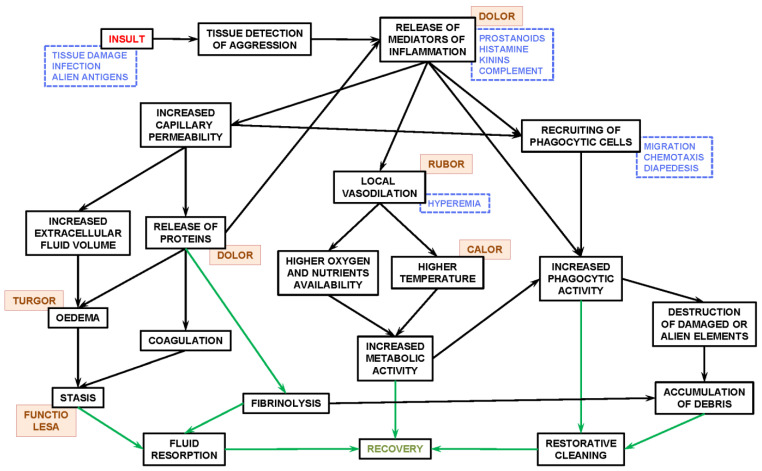
**Gross mechanisms of ‘Clinical Inflammation’**. The inflammatory processes are shown as black rectangles linked by arrows. Brown attached labels show the correspondences with Celsus-defined components of classic inflammation. Blue attached rectangles explain the agents and mechanisms responsible for the described effect. The actions developed during the recovery phase (i.e., after ceasing the inflammatory phase) are shown as green arrows.

**Figure 4 ijms-25-02251-f004:**
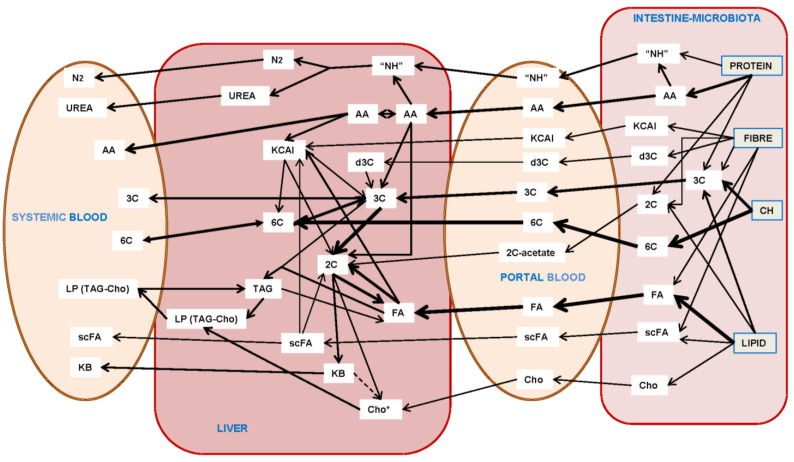
**Shift in the composition of digested food in the intestinal lumen to systemic blood due to the combined triage of molecules carried out by the intestine and liver.** The main classes of nutrients present in the liver of a human omnivore (protein, fiber, carbohydrates [CH], and lipid). The arrows’ sizes represent the relative relevance of a number of processes related to the usual composition of human diets. The processes shown require a ‘signal-armored’ vein (the porta) because of the presence of bioactive compounds and usually high concentrations of common metabolites. This load of compounds requires homeostatic regulation of the liver (via modification, catabolism, and dilution), as well as taking-up, storing, or metabolizing compounds (when already in the partition mode) to minimize changes, as best as possible, iwithin the physiological range in the levels and distribution of systemic blood nutrients. Abbreviations used (from left intestine to right, and from the highest line to lowest): “NH” = small N-containing molecules, such as ammonia; AA = amino acid; KCAIs = Krebs cycle anaplerotic intermediate catabolites of amino acid metabolism (Figure 5); d3C = 3-hydrocarbon fragments with D conformation, such as D-lactate; 3C = 3-hydrocarbon fragments, with gluconeogenic potential, such as L-lactate, pyruvate, glycerol, alanine, serine, etc.; 2C = 2-carbon fragments (derived from 3C oxidative decarboxylation or generated in the β-oxidation of fatty acids, i.e., acetyl-CoA or acetate); 6C = 6-carbon carbohydrate units, i.e., hexoses, largely glucose ((or fructose, galactose) but also pentoses, obviously containing 5C, but catabolically restructured to 3C or 6C via the pentose-P pathway and confluent pathways); FA = fatty acids, essentially formed by the sequential incorporation of 2C units and oxidized via β-oxidation; scFA = short-chain fatty acids, such as propionate (a KCAI, but not in fact a canonical 3C) and butyrate, largely microbiota products of high regulatory importance; Cho = cholesterol (and some of its derivatives, essentially sterols); N_2_ = nitrogen gas, the ultimate excreta of the ‘alternative’ amino-disposal pathway; UREA = urea; TAGs = triacylglycerols; KB = ‘ketone bodies’, i.e., 3OH-butyrate, acetoacetate (and acetone); Cho* = cholesterol, including both dietary (i.e., that brought in with the portal blood) and that synthesized in the liver from 2C via the mevalonate pathway; LP(TAG-Cho) = lipoproteins, mainly VLDL, containing TAGs and cholesterol.

**Figure 7 ijms-25-02251-f007:**
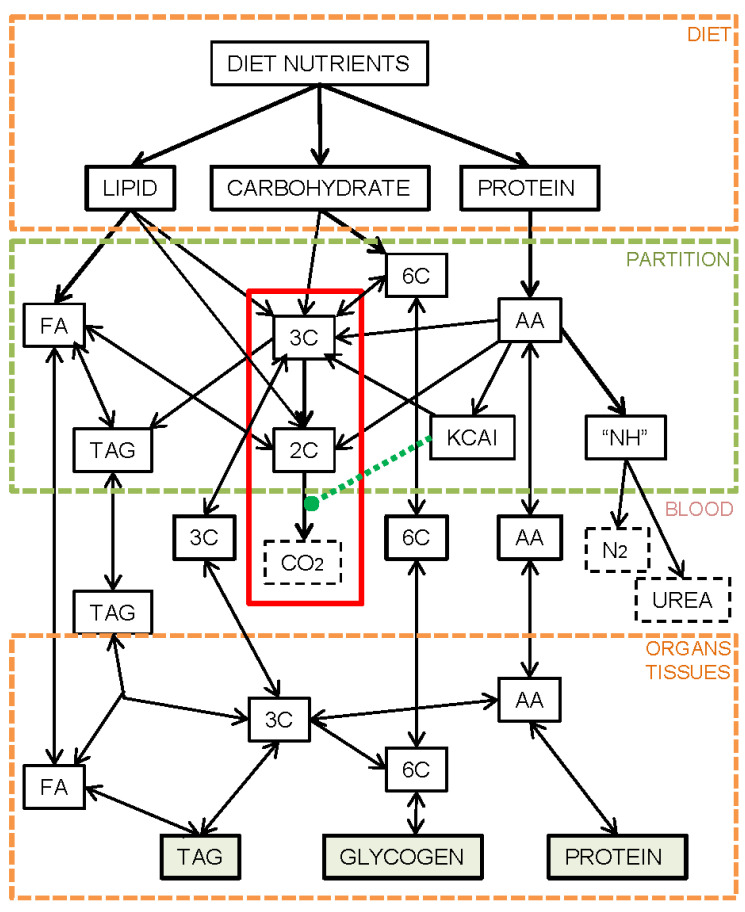
**Simplified representation of the process of assimilation of nutrients, substrates partition, and their relationship with organ metabolism and body energy reserves.** The core of partition is the programmed efficient use of available amino acids (AA), fatty acids (FAs), and carbohydrate units (6C) to fuel the tissues’ needs via the breakup and oxidative catabolism of (a) TAG and FA, (b) 6C and 3C, and (c) protein-derived amino acids. This is complemented by the elimination of catabolites (CO_2_, urea, N_2_, and water, *not depicted in the graph*). The essential path for the generation of energy from the available substrates is marked with a red border: the transition of 3C (i.e., lactate/pyruvate) to 2C (acetyl CoA, via pyruvate dehydrogenase complex) and, finally, to CO_2_ (through the whole Krebs cycle) inside the mitochondria. In this graph, this process represents the metabolic changes taking place in the liver, as the first recipient of the nutrient load discharged from the gut, but this mechanism is mimicked (with a few variations) by almost every other tissue/organ/cell of the body. The necessary conjoint catabolism of amino acids and that of 3C/2C units in the mitochondria is first facilitated by the anaplerotic effect of KCAI (dotted green line) and is a highly regulated and resilient system by which most of our energy needs are covered.

**Figure 8 ijms-25-02251-f008:**
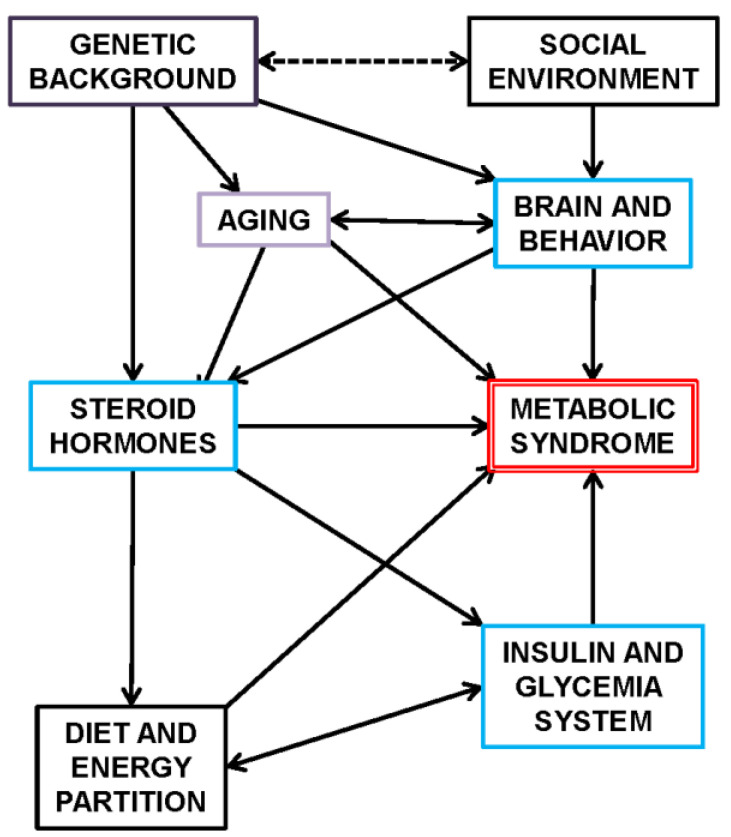
**Main factors implied in the development of MS**.

**Table 1 ijms-25-02251-t001:** **Main symptoms/traits described as part of the MS cluster of disorders/diseases**.

MS Main Specific Pathology Family	MS-Related Agent, Effect, Action, or Pathology	Organ/System Implicated	Comments
** *Genetic/epigenetic* ** * **predisposition factors** *	*monogenic and polygenic obesity* *diabetes and other disorders*	all systems, adipose depots, energy balance-systems failure	gene-defective signals/altered systems
	genetic predisposition to cancer	reproductive, interstitial, other	gene-related cancer
	inherited predisposition to system or organ failure	cardiovascular, liver, kidney, hematopoietic, interstitial, other	defective gene-related alterations
**Nervous system(s)**	bulimic/anorexic disorders	nervous and endocrine systems	appetite + behavioral
	*depression*		nervous signal disorder
	hypothalamic disorders		adrenals/gonads
	*parasympathetic predominance*		phased-out (?) affectation
	*neuropathy (central, peripheral,* *sensory), chemical senses, glaucoma*	nerves from central, autonomic and enteric nervous systems, sensory systems	wide range of peripheral- and senses affectation
**Muscle and nervous** **signal disorders**	*fibrillation, disordered heart rhythm*	heart, muscle and circulatory, brain	
* **Altered energy** * ** *expenditure* **	*sedentary behaviors, lack of* *exercise*	muscle, skeleton, splanchnic bed, interstitial	essentially related to insufficient energy expenditure with respect to intake, also psychogenic causes
	immobilization, extreme rest	muscle, skeleton, brain	bed rest, paralysis, etc.
	body heat and thermogenesis-related disorders	liver, brown adipose tissue, muscle, brain	insufficient thermogenic capability or regulation. Hyperthermia/fever
**Mitochondrial** **function disorders**	mitochondrial cycle failure	liver, nervous system	
**Rheumatic-related** **disorders (immune** **and *inflammatory*)**	rheumatoid arthritis (and *osteoarthritis*)	interstitial, skeletal, intestine, nervous, all systems	additional large and varied pathologic relationships
	altered redox states and purine metabolism, *hyperuricemia, gout*	interstitial, all systems	
	atopia, *psoriasis*	skin, intestine, nervous, immune	relationships with cancer and other MS conditions (e.g., obesity)
* **Diet; excess energy** * * **intake *vs.* losses** * * **(and nutrient/energy** * * **triage/partition)** *	*excess energy intake* *(uncompensated)*	liver, kidney, muscle, all systems	includes the dietary protein paradox
	*liver steatosis* (and fibrosis)	liver, circulatory, brain, interstitial	probably a primordial cause of MS
	*relative excess of carbohydrate*	liver, muscle, interstitial, endocrine	
	insufficient (relative) N intake	liver, kidney, muscle, all systems	includes the dietary protein conundrum
	*obesogenic (lipid, sugars) diets*	liver, interstitial, endocrine	
	*excess energy available and ectopic* *deposition (obesity)*	liver, muscle, interstitial, skin, all systems	a main cause of obesity
	*purified/ultra-processed diets*	liver, muscle, interstitial, endocrine	probably a mix of relative excess of carbohydrate and obesogenic diets
	essential nutrient deficits	liver, nervous system, all systems	deficits affecting regulation
** *Excess body lipids* **	*obesity*, all causes	interstitial, all systems	*relative and absolute excess of lipid in* *the body including ectopic depots*
	*altered adipose metabolic* *control-related depots*	heart and vessels-circulatory control-related depots	control of heart and vessel blood dynamics
	*disordered tissue fat deposition*	interstitial, muscle, all systems	fat depots for direct organ use
	*hyperlipidemia*	liver, peripheral organs/tissues, blood	*hypertriacylglycerolemia,* *hypercholesterolemia, and* *apolipoprotein-related disorders*
** *Secondary to excess* ** * **body lipids** *	*atherosclerosis*	circulatory, liver, blood vessels	altered blood flow and heart overburden
	*arterial hypertension*	circulatory, kidney, brain, liver, other	vessel endothelial damage, increased blood flow resistance,
	*body mass-related respiratory and* *hemodynamic disorders*	heart-circulatory, interstitial, respiratory, nervous, all systems	disordered supply of oxygen and nutrients, altered blood flow and renal control of water and electrolytes. Hypoxia
	hypotension/heart insufficiency		*chronic hypoxia, edema, lymphedema,* *inflammation*
	*obesity-related cancer*	liver, other splanchnic, nervous, all systems	fibroblasts and adipocytes as necessary energy support for cancer types
**Toxic-induced disorders**	chronic and acute toxicity	liver, interstitial, all systems	includes alcohol and other drugs and iatrogenesis (i.e., thyroid)
	carcinogen-induced cancer	liver, interstitial, brain, reproductive system	not part of MS but both/either favoring MS or facilitated by MS
** *Insulin and diabetes* ** * **mellitus** *	*prediabetes*	all systems, liver	
	*insulin resistance*	insulin function, all organs	a main recognized factor of MS
	gestational diabetes	all systems, liver, placenta	temporary
	*type 2 diabetes*	pancreas, liver, interstitial, all systems	*close relationship with obesity and* *other endocrine disorders of energy* *handling*
	polycystic ovary syndrome	ovary, adrenals, liver, all systems	
**Other hormone disorders**	hypothyroidism	thyroid, interstitial, liver, all systems	
	hypercortisolism	adrenals, brain, endocrine, interstitial, circulatory, all systems	often not defined within MS but as separate entities
	*testosterone deficit*, andropause	testicle, muscle, bone, liver, endocrine, brain, all systems	a main factor in developing MS
	*DHEA deficit*	adrenals, brain, liver, reproductive, all systems	
	*estrogen deficit*, menopause	adrenals, reproductive, liver, interstitial, all systems	related to testosterone deficit
	altered handling of water/ions	adrenals, kidney, brain, circulatory, all systems	unclear association with BW; ion handling effects on diet or metabolic regulation
	*hormone-related cancer*	reproductive, endocrine, all systems	*ovary, endometrial, breast, prostate* *cancers as main examples*
**Sleep apnea**	so far unknown, albeit syndromic with obesity and inflammation	cartilage, respiratory, circulatory	mechanism of pathogenesis not yet related to MS, but syndromic with obesity
**Microbiota-related** **disorders**	altered microbiota states	microbiota, intestine, liver, interstitial, nervous, skin, all systems	loss of digestive efficiency; infection and forfeiting of the host-biota immune status quo
	interspecies gene transfer	all systems	integration of viral and bacterial genes into the human genome, with quite different long-term effects (part of evolution?)
**Infection**	virus infection and artificial gene transfer	immune, interstitial, all systems	virus-transmitted obesity
	parasites, sepsis	immune, intestine, interstitial, all systems	MS as a sequel or consequence of infection/recovery or overreaction

In *italics*, ‘classical’ or commonly cited symptoms, disorders, and diseases included in the clinical concept of MS.

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
