# Peer review of "The Metabolic Syndrome, a Human Disease"

_ijms, 2024, doi:10.3390/ijms25042251_

Round 1
Reviewer 1 Report
Comments and Suggestions for Authors
Review article titled (The metabolic syndrome, a human disease) by Alemany discussed the metabolic syndrome as a human disease starting from definitions, nature, to pathology and symptoms. The review contains huge number of references that is considered unpractical and not common. I find this review very hardly to be comprehended and not useful at all, contains old info and not interestingly or correctly written.
1- Title is too brief and does not give the full idea about the content, need to be revised to adequately express the content of the paper.
2- The use of the abbreviation "MS" is not common and widely used for "multiple sclerosis", I wish if authors can find a better alternative for this abbreviations.
3- how figure 1 was created? If certain software or database was used it will be worth to mention it.
4- Figure 1 is very difficult to be comprehended by the reader, simpler figure or multiple smaller figures will be more useful
Also figure 2 & 4 & 5 & 7
5- The number of references should be reduced to a maximum of 100 reference. How authors proposed to use all these references?
6- Abstract: exceeds the number of word limit for the journal, kindly check the guide for authors
7- Key words: what is "adipose organ"?
8- Key words: authors used non useful key words; they should use words that best serve the indexing of this paper in the future.
Also the word limit exceeds the journal guidelines
9- Intro : authors used a title "The human nature of MS" this does not make sense!! I think athors mean 'nature of human MS"
10- Line 53-56: do authors find these info related to the topic?
11- In general, the review can be shortened to be more concrete by removing the unrelated parts
12- Line 213: 2.2. MS main paths and mechanisms: this does not make sense
13- Table 1 has no references !!! All these huge amount of info without references?
14- Line 789: "6. The MS disease" what authors wish to describe here?
15- Table 2 contains huge amount of information without refernces; such as TABLE 1
16- Fifty pages about MS deserves to be a book chapter (after adequate rewriting) not a literature review
Comments on the Quality of English LanguageNeeds extensive revision
Author Response
See attached pdf file including the response to Reviewer

Reviewer 2 Report
Comments and Suggestions for Authors
The article is an extensive review of a relevant topic: metabolic syndrome. Several points are covered with many references on the subject.
However, although the article is up to date and complete, it is very long and may compromise the reader's attention to finish reading.
Some topics can be summarized, as they are covered in more than one point in the article, thus reducing the size of the document.
As the text is very long, including a brief main message at the end of each topic would make it easier to consolidate the knowledge of that specific part.
In topic 4, make correlations between the physiology and pathophysiology of MS. Please include more references to 4.2. topic.
To develop MS, is it more critical to have visceral fat or not to have peripheral fat? Could you comment on lipodystrophies?
Author Response
See the attached pdf file with my response to Reviewer

Reviewer 3 Report
Comments and Suggestions for Authors
In this manuscript, the authors reviewed the latest research outcomes on metabolic syndrome. They discussed MS triggering and maintenance factors, with especial emphasis on inflammation. They also addressed the metabolic regulation by energy balance and dietary energy assimilation, triage and partition. In general, this review will contribute to the study on metabolic syndrome by improving the understanding of MS paths and mechanisms.
Before the final acceptance, I will list a few issues to be addressed in the revised manuscript.
1. Authors particularly focused on inflammation in this review and almost spent half of pages to discuss the roles and regulation of inflammation in MS. This point should be strengthened in their abstract.
2. There are too many arrows in Fig 4, 7, and 8, which make it difficult to understand. Some rearrangements on those figures are needed.
Comments on the Quality of English LanguageThe English looks good in current version of the manuscript.
Author Response

(The authors gave the same response as above.)

Reviewer 4 Report
Comments and Suggestions for Authors
the article is very complete and well written. I think it's an important summary of MS. However, I think it contains too many references (more than 1000) and there's no need for this number in most of the sentences. The images are very important, but not very visually appealing, I think they could be much better. The tables are informative, but there are too many colors in the text, and in the last one we no longer know what each color represents, so it could have another, clearer visual appearance. However, I really liked the article and I think it should be accepted.
Author Response
See the attached pdf file with my response

Round 2
Reviewer 1 Report
Comments and Suggestions for Authors
The revised version of paper titled (The metabolic syndrome, a human disease) was not adequately prepared and still huge and unclear and not easily readible. Authors gave huge replies in the reply letter without value and kept the manuscript unrevised. I regret I cannot recommend it for publication.
Comments on the Quality of English Languagefine
Author Response
Please, see the attached document